# Hydroxymethanesulfonate formation accelerated at the air-water interface by synergistic enthalpy-entropy effects

Jifan Li[1,5], Weiqiang Tang [2,5], Jiabao Zhu[1], Jinrong Yang [1] ✉ & Xiao He [1,3,4] ✉

Hydroxymethanesulfonate is a key organosulfate linked to severe fine-particle pollution in fog and clouds, yet its rapid formation mechanism at the air-water interface remains elusive. Here, using metadynamics-biased ab initio molecular dynamics simulations, high-level quantum chemical calculations and reaction density functional theory, we reveal that synergistic enthalpy-entropy effects govern the nucleophilic addition between bisulfite and formaldehyde. Compared to the gaseous reaction, the aqueous reaction faces a ~5.0 kcal/mol water reorganization barrier, partly offset by polarization effects. Ab initio molecular dynamics simulations show hydrogen bonding networks facilitate proton transfer via the Grotthuss mechanism, reducing activation entropy by ~5.5 kcal/mol. At the interface, partial solvation and restricted formaldehyde motion lower the enthalpy and configurational entropy by ~1.0 and ~0.9 kcal/mol, respectively, alongside a 1.9 kcal/mol electric field effect. These combined effects enhance the interfacial reaction rate by two orders of magnitude, offering insights into heterogeneous chemistry and strategies for winter haze mitigation.

Atmospheric aerosols are integral to the Earth's system, exerting profound influence on ecosystem productivity, human health, horizontal visibility, and severe haze pollution. Among the various types of atmospheric aerosols, organic aerosol constitutes 20-90% of the mass fraction in submicrometer aerosols[1,2], with secondary organic aerosol dominating globally[2-4], contributing to 76% of ambient organic aerosols[5]. Hydroxymethanesulfonate (HMS, $HOCH_2SO_3^-$) not only acts as a potential S(IV) reservoir in secondary organic aerosols[6-9] but also can undergo oxidation to form sulfuric acid[10-15], then influencing atmospheric chemistry. Historically, HMS concentrations in atmospheric aerosols were found to be low in regions, such as United States, Germany, and Japan, partially due to the misidentification of organosulfur compounds as inorganic sulfate in conventional

measurements[16-20]. Recently, the elevated HMS concentrations, particularly during wintertime in north China[6,9,12,16,21-25] and arctic areas[18,19,26-29], have been detected by advanced measurement techniques, including ultra-high performance liquid chromatography orbitrap mass spectrometry[16], aerosol time-of-flight mass spectrometer[30], particle-into-liquid sampler-ion chromatography[19] system, and Fourier-transform infrared spectroscopy[31]. For example, using the particle-into-liquid sampler-ion chromatography system, Campbell et al.[18] found that the average concentration of 2020 in wintertime Fairbanks reached to 2.4 µg/m³ when PM$_{2.5}$ exceeded 35 µg/m³. In addition, Wei et al.[16] reported a peak HMS concentration 7.3 µg/m³ in wintertime Beijing by using ultra-high performance liquid chromatography orbitrap mass spectrometry. These field observations highlight

[1]Shanghai Engineering Research Center of Molecular Therapeutics and New Drug Development, Shanghai Frontiers Science Center of Molecule Intelligent Syntheses, School of Chemistry and Molecular Engineering, East China Normal University, Shanghai 200062, China. [2]School of Chemical Engineering, East China University of Science and Technology, Shanghai 200237, China. [3]Chongqing Key Laboratory of Precision Optics, Chongqing Institute of East China Normal University, Chongqing 401120, China. [4]New York University–East China Normal University Center for Computational Chemistry, New York University Shanghai, Shanghai 200062, China. [5]These authors contributed equally: Jifan Li, Weiqiang Tang. ✉e-mail: jryang@chem.ecnu.edu.cn; xiaohe@phy.ecnu.edu.cn

the importance of elucidating the formation mechanism of HMS as a key aspect of understanding secondary organic aerosol production and developing effective haze control strategies[6,32].

The chemistry of HMS formation was first explored in laboratory studies during the 1980s[27], providing fundamental evidence for the pivotal role of aqueous pH in HMS formation[33,34]. For instance, Purnendu et al.[33] used UV-vis spectrum to determine the equilibrium constant $K$ for the reaction HCHO + S(IV) $\rightleftharpoons$ HMS under different pH, with $1/K$ value of $1.17 \times 10^{-5}$ M at pH 5 and $2.53 \times 10^{-5}$ M at pH 6. They further proposed that HMS was directly formed through the nucleophilic addition reaction between bisulfite ($HOSO_2^-$) and formaldehyde (HCHO). Moreover, the spectrophotometric kinetic study by Boyce et al.[35] quantified the aqueous forward reaction rate constant for $HOSO_2^-$ (aq.) + HCHO(aq.) $\rightarrow$ $HOCH_2SO_3^-$ and $SO_3^{2-}$ (aq.) + HCHO(aq.) $\rightarrow$ $O^-CH_2SO_3^-$ reaction as $7.90 \pm 0.32 \times 10^2 M^{-1} \cdot s^{-1}$ and $2.48 \pm 0.05 \times 10^7 M^{-1} \cdot s^{-1}$, respectively. Meanwhile, many field measurements pointed out that the atmospheric pH during HMS formation is primarily within a range of 2–7[7,8,36–38]. Although some field observations found that HMS concentration reaches maximum at the pH range of 4–6[7,36,37,39], several field studies also reported excess HMS under more acidic environment in fog and cloud water[32,37,38,40]. For example, Huang et al.[40] documented rapid formation of HMS in the aerosol pH range of 3.3–4.3 during January in north China, where the HMS-containing particle number can constitute 50% of the total particle number. In this atmospheric acidic condition, the dissolved S(IV) exists as the tautomeric equilibrium between bisulfite ($HOSO_2^-$) and sulfonate ($HSO_3^-$)[32,37,38,40–42]. Therefore, this poses a contradiction that arises from the existing high concentration of HMS in the acidic aqueous environment with the abundant $HOSO_2^-$, because the reaction rate of HCHO with $HOSO_2^-$ in bulk solution is five orders of magnitude lower than that with $SO_3^{2-}$.

Numerous experimental investigations have revealed that the role of HMS formation in heterogeneous environment cannot be negligible[9,18,24,43–45]. Several studies have emphasized the high value of Henry's law constant of HCHO[18,27,37,46], which reaches to $1.8 \times 10^5$ M/atm at 273 K observed by Campbell et al.[18]. Additionally, Zhang et al.[47] demonstrated that the HMS formation rate in aerosol water at an ionic strength of 3.2 M was 1.5 folds higher than that in bulk water under the smog chamber system. Recent field observations further indicated that HCHO could accumulate at the air-water interface[8,38,44,48,49], corroborated by the vibrational sum frequency spectroscopy[50,51]. Ota et al.[50] pointed out that the $SO_2$–HMS complex would enhance the surface concentration upon exposure to $SO_2$. Theoretical studies have also calculated the reaction energy barrier for gaseous HMS formation pathway[52], confirming the more effective formation of $SO_3^{2-}$ + HCHO $\rightarrow$ $O^-CH_2SO_3^-$ reaction. Furthermore, Chen et al.[53] introduced a single water molecule as a participant of the $HOSO_2^-$ (g.) + HCHO(g.) reaction with a lower reaction energy barrier, identifying the non-negligible role of water participation. However, current theoretical investigations have not accounted for the influence of atmospheric heterogenous environment, resulting in a substantial gap between the field measured elevated HMS concentrations under cloud or fog environment and the traditional atmospheric scenario that merely hinges on aqueous or gaseous reaction pathway. Furthermore, molecular dynamics simulation of the interface vibrational sum frequency spectroscopy[54,55] showed the interfacial constraint imposed on the tilt angle of reactant, which sheds light on the emerging mechanism of the HMS formation at the air-water interface. Yet, these studies are grounded solely in the aqueous and gas phase reaction of HMS formation, leaving the underlying chemistry mechanism of heterogeneous HMS formation poorly understood.

Herein, we combine metadynamics-biased ab initio molecular dynamics (AIMD) simulations with high-level quantum chemistry calculations and reaction density functional theory[56–58] (RxDFT) to quantitatively decode the mechanisms underlying the rapid HMS formation by the nucleophilic addition between $HOSO_2^-$/$SO_3^{2-}$ and HCHO

at the air-water interface. Our findings indicate that the heterogeneous HMS formation is predominantly governed by the synergistic regulation of entropy and enthalpy, rather than the isolated contributions of interfacial electric field or partial solvation. The RxDFT analysis reveals that, compared to the gaseous HMS formation under acidic condition, the reactants in the homogeneous bulk solution should overcome a water reorganization barrier of ~5.0 kcal/mol, partially mitigated by the water polarization effect (~2.1 kcal/mol). Interestingly, according to the AIMD simulation, the water-mediated proton transfer through Grotthuss mechanism is found to reduce the activation entropy by ~5.5 kcal/mol relative to intramolecular proton transfer in gas phase, thereby lowering the free-energy barrier for HMS formation to ~11.4 kcal/mol, consistent with the experimental value of 13.6 kcal/mol within the margin of error. Furthermore, the partial solvated HCHO at the air-water interface reduces the activation enthalpy by ~1.0 kcal/mol, and the interfacial stabilization effect arising from the frustrated translation and rotation of HCHO decreases the configurational entropy barrier by ~0.9 kcal/mol. This synergistic regulation of entropy and enthalpy effect is comparable to interfacial electric field effect with a reduction of 1.9 kcal/mol. Collectively, these effects result in approximately two orders of magnitude enhancement in heterogeneous reaction rate compared to the aqueous reaction. Meanwhile, the free-energy profile of the HMS formation shows that the calculated energy barrier of $SO_3^{2-}$ (aq.) + HCHO(aq.) $\rightarrow$ $O^-CH_2SO_3^-$ reaction in aqueous solution is ~5.8 kcal/mol, which is close to the experimental result (7.1 kcal/mol) within the relative error margin. For the corresponding heterogeneous reaction, the energy barrier at the air-water interface is reduced to ~2.7 kcal/mol, resulting in approximately hundred-fold increase in reaction rate relative to the aqueous phase. These results provide quantitative insights into the mechanisms of HMS formation under different pH conditions, advancing our understanding of atmospheric chemistry and informing strategies for effective air quality management. Furthermore, the computational pipeline used in this study offers a mechanistic framework for exploring heterogeneous reactions at gas-water interfaces, potentially accelerating the large-scale microdroplet technologies for green synthesis.

## Results

### Overall glance

The nucleophilic addition mechanism between $HOSO_2^-$/$SO_3^{2-}$ and HCHO has been previously proposed by Purnendu et al.[33] and Boyce et al.[35]. In this mechanism (Fig. 1a), $HOSO_2^-$/$SO_3^{2-}$ functions as a nucleophile and HCHO functions as an electrophile. The initial nucleophilic attack results in the formation of a C–S single bond, and upon reaction between HCHO and $HOSO_2^-$, a proton transfer process subsequently yields the hydroxyl group. However, current theoretical investigations[52,53] have not accounted for the influence of atmospheric heterogenous environment, resulting in a substantial gap between the elevated HMS concentrations under cloud or fog environment in field measurements and the traditional atmospheric scenario that hinges on aqueous or gaseous reaction pathway (Fig. 1b), particularly at acidic condition. This study aims to bridge this gap by employing AIMD simulations and to explore the intricate mechanism of HMS formation (Fig. 1c).

### HMS formation under acidic condition

Recent atmospheric field measurements have reported elevated HMS concentration in fog and cloud water under acidic conditions, where the ratio of S(IV) species is dependent on the tautomeric equilibrium between $HOSO_2^-$ and $HSO_3^-$[32,37,38,40–42]. We first examined the nucleophilic reaction of $HOSO_2^-$ and HCHO as a typical pathway for HMS formation. As delineated in Fig. 2a, this reaction initiates the nucleophilic attack followed by the proton transfer. In the first step, the lone-pair of electrons on the sulfur atom of $HOSO_2^-$ attacks the

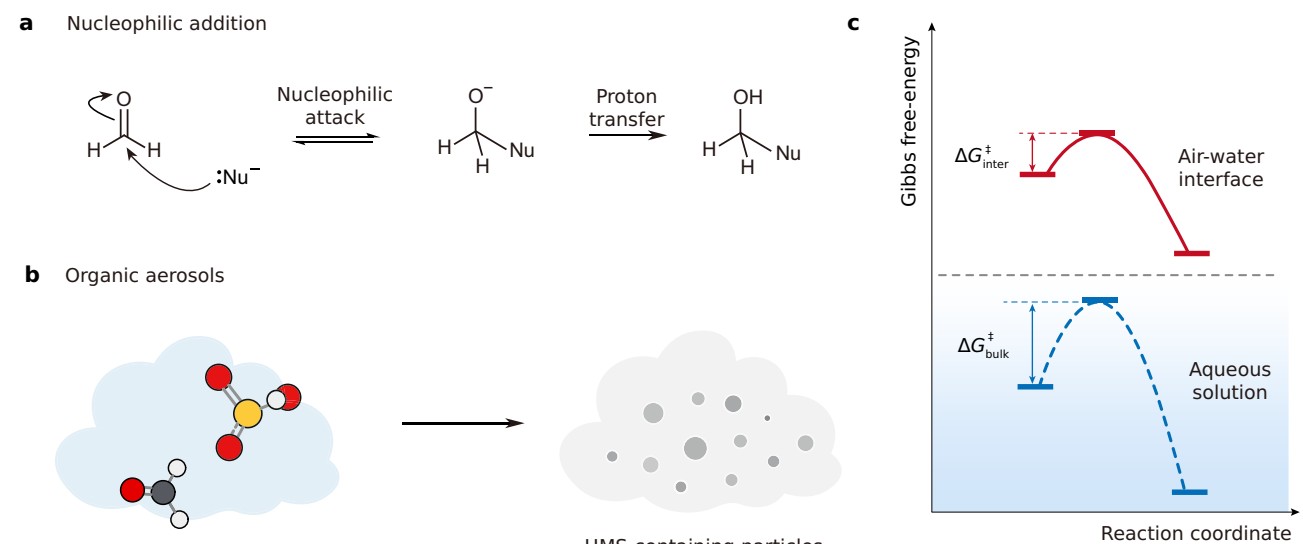

**Fig. 1 | Air-water interface as accelerator for HMS formation. a** A representative reaction mechanism of nucleophilic addition for formaldehyde. $HOSO_2^-/SO_3^{2-}$ functions as a nucleophile ($Nu^-$) and HCHO functions as an electrophile. The initial nucleophilic attack results in the formation of a C–Nu single bond and a proton transfer process subsequently yields the hydroxyl group. **b** The HMS-containing particles (right) is generated by the reaction between HCHO and $HOSO_2^-$ (left) in acidic organic aerosols. **c** Comparison for the reaction energy barrier between aqueous formation ($\Delta G_{bulk}^{\ddagger}$, blue dash line) and heterogeneous formation ($\Delta G_{inter}^{\ddagger}$, red line).

carbonyl carbon atom of HCHO, leading to a C–S bond formation in a tetrahedral intermediate. Subsequently, HMS is generated with a proton transfer process. As shown in Fig. 2b, the gaseous reaction needs to overcome a relatively high free-energy barrier of $\Delta G_{gas}^{\ddagger}$ (14.0 kcal/mol). For transition state (**TS**), $HOSO_2^-$ and HCHO form a five-membered ring via hydrogen bonding [H($HOSO_2^-$)···O(HCHO)] and nucleophilic interaction [S($HOSO_2^-$)···C(HCHO)], which triggers the intramolecular proton transfer from the hydroxyl of $HOSO_2^-$ to the carbonyl oxygen of HCHO.

The free-energy profiles of HMS formation in bulk solution and at the air-water interface were obtained by three independent metadynamics-biased AIMD simulations. The collective variable (CV) was defined as the distance between the sulfur atom of $HOSO_2^-$ and the carbon atom of HCHO, as shown in Fig. 2c. The corresponding evolution of CV as a function of time for each three independent simulations are presented in Supplementary Figs. 4 and 5. Although the metadynamics simulations likely did not reach convergence due to the lack of recrossing events, the resulting free energy profiles and corresponding conclusions have been supported by thermodynamic integration (TI)-AIMD simulation. The average free-energy barrier for the aqueous reaction of HMS formation ($\Delta G_{bulk}^{\ddagger}$) is determined to be 11.4 ± 0.8 kcal/mol, which is close to the experimental result ($\Delta G_{exp}^{\ddagger}$ = 13.6 kcal/mol) reported by Boyce et al.[35]. The air-water interface leads to the corresponding free-energy barrier reduced to 7.6 ± 0.6 kcal/mol (Fig. 2e), which is basically impregnable within an appropriate range in the larger system simulation (see Supplementary Fig. 6). According to the rate equation proposed by Boyce et al.[35], the rate equation of nucleophilic addition can be expressed as a second-order reaction:

$$\nu = k[HOSO_2^-][HCHO] \tag{1}$$

Here, $\nu$ is reaction rate, $k$ is the rate constant, and $[HOSO_2^-]$ and [HCHO] represent the concentration of $HOSO_2^-$ and HCHO, respectively. Combined with the Eyring equation:

$$k = \frac{\kappa k_B T}{h} e^{-\frac{\Delta G^{\ddagger}}{RT}} \tag{2}$$

where the rate constant is determined by the activation energy $\Delta G^{\ddagger}$, $\kappa$ is the transmission coefficient, $k_B$ is the Boltzmann constant, $T$ is the temperature, R is the universal gas constant, and h is the Planck constant. The ratio of the rate constant between interface ($k_{inter}$) and aqueous ($k_{bulk}$) reaction can be inferred as:

$$\frac{k_{inter}}{k_{bulk}} = e^{\frac{\Delta G_{bulk}^{\ddagger} - \Delta G_{inter}^{\ddagger}}{RT}} \tag{3}$$

By substituting the $\Delta G_{bulk}^{\ddagger} - \Delta G_{inter}^{\ddagger}$ = 3.8 kcal/mol and R$T$ = 0.6 kcal/mol, we find that the reaction rate at the air-water interface is ~563 times faster than that of the aqueous formation. This implies that the non-negligeable impact of $HOSO_2^-$ with a lower energy barrier of heterogeneous process is sufficient to generate abundant HMS at acidic condition under aqueous aerosol environment.

The snapshot structures of HCHO + $HOSO_2^-$ reaction in bulk solution and at the air-water interface are depicted in Fig. 2d and f, respectively, with the corresponding CV changes over time delineated below each snapshot. In the initial stages of the simulations for 0–50 ps in bulk solution and 0–75 ps at the air-water interface, the CVs both fluctuate within the range of 3–5 Å. Subsequently, $HOSO_2^-$ approaches to HCHO, initiating the nucleophilic attack on the carbonyl group of HCHO and progressing towards the transition state. Simultaneously, HMS formation is facilitated through a water-participated tautomerization, where the hydrogen-bonding networks of water molecules (ball type) act as the proton transfer channel. The proton transfer loop mechanism in the aqueous reaction involves a five-membered ring composed of the oxygen atoms of three water molecules, the hydroxyl oxygen and the carbonyl oxygen, enabling the proton at the hydroxyl oxygen of $HOSO_2^-$ transferred to the carbonyl oxygen of HCHO. At the air-water interface, the proton transfer occurs through a six-water molecule network situated at the surface layer.

The structural analysis of the air-water interfacial reaction is depicted in Fig. 3a to c, with complementary analysis of aqueous reaction shown in Supplementary Fig. 11. Figure 3a tracks the center-of-mass (COM) of HCHO (red line) and $HOSO_2^-$ (blue line) along the z-direction over simulation time with alongside the average density (grey region). The dashed grey horizon line at z = 10.4 Å represents the

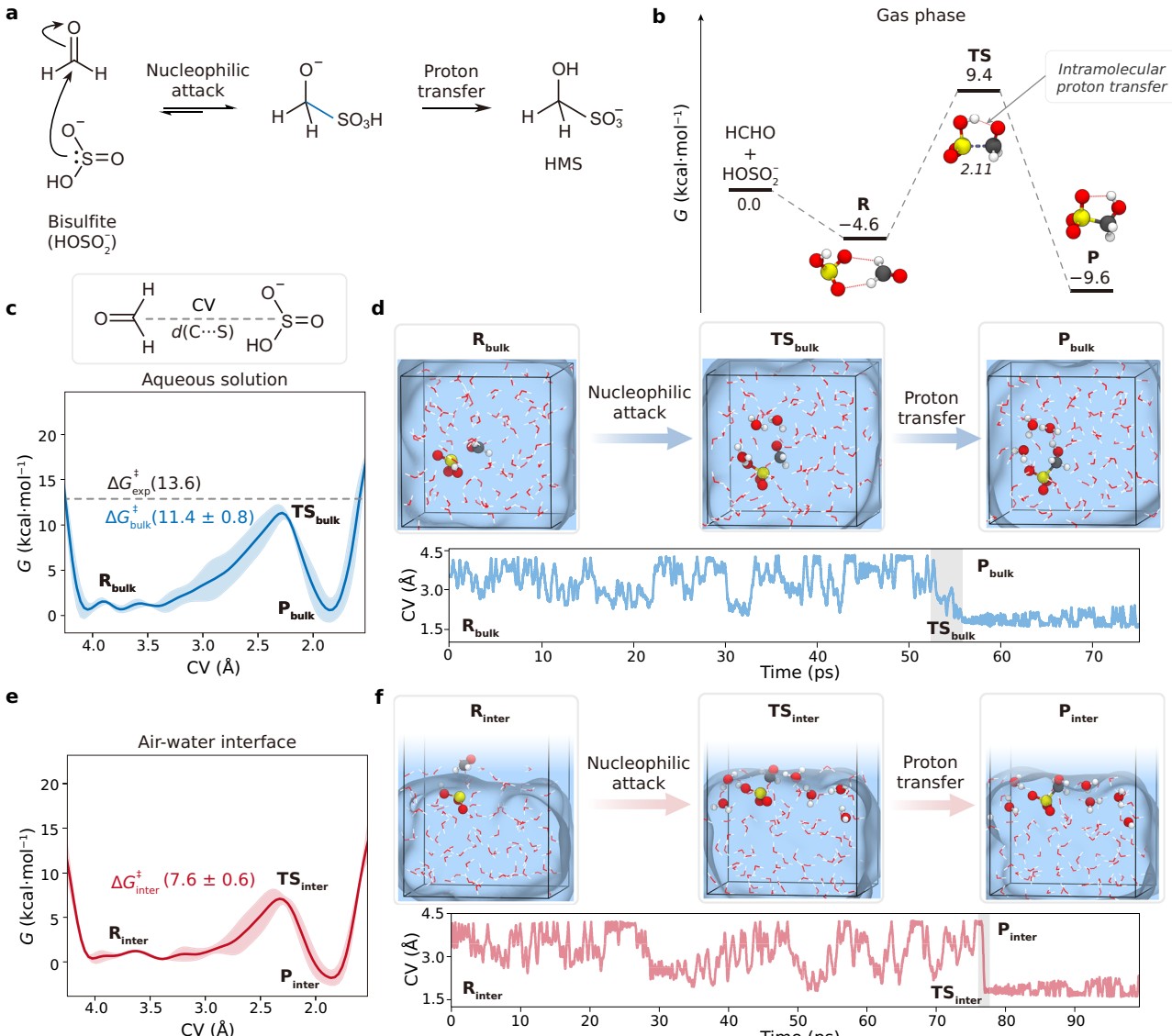

**Fig. 2 | HMS formation under acidic conditions. a** A representative mechanism for HCHO + HOSO$_2^-$ reaction. **b** Gibbs free-energy profiles for HCHO + HOSO$_2^-$ reaction in gas phase and their corresponding structures of the stationary points optimized at M06-2x/6-311++G(d, p) level of theory. Relative single point energies of reactant complex (**R**), transition state (**TS**), and product (**P**) were obtained at CCSD(T)/aug-cc-pVTZ level of theory. **c** (Top) Schematic representation of the CV. (Bottom) Gibbs free-energy profile (blue line) with error band (shaded area) for aqueous HCHO + HOSO$_2^-$ reaction compared with the experimental result ($\Delta G_{exp}^{\ddagger}$) of Boyce et al.[35]. Relevant error band is calculated on the standard deviation by employing cubic interpolation on three independently obtained free-energy profiles. The error of the free-energy barrier is the standard deviation of the free-energy barriers of three metadynamics simulations. **d** (Top) Snapshot structures (reactant, **R**$_{bulk}$, transition state, **TS**$_{bulk}$, and product, **P**$_{bulk}$) obtained from a single metadynamics-biased AIMD simulations. (Bottom) Relevant CV variation as a function of the simulation time for aqueous reaction (blue line). **e** Gibbs free-energy profile (red line) with error band (shaded area) for the heterogeneous HCHO + HOSO$_2^-$ reaction. Calculation of error band is same as the bulk reaction simulation. **f** (Top) Snapshot structures (**R**$_{inter}$, **TS**$_{inter}$, and **P**$_{inter}$) obtained from a single metadynamics-biased AIMD simulations. (Bottom) Relevant CV variation as a function of the simulation time for interfacial reaction (red line).

interface boundary. Throughout the reaction, HCHO resides at the outmost surface, while HOSO$_2^-$ is located at the subsurface region. After HMS formation, the product of HMS gradually migrated into the bulk water, aligning with the vibrational sum frequency spectroscopy observation by Ota et al.[50]. A comparison of the interaction free-energy reveals that the interaction between HMS and HOSO$_2^-$ (−2.0 kcal/mol) is stronger than that between HMS and SO$_2$ (1.2 kcal/mol) (Supplementary Fig. 21 and Supplementary Table 4). This suggests that the dissolved SO$_2$ at the air-water interface forms HOSO$_2^-$, which subsequently binds with the interfacial HMS to generate a complex, enhancing the HMS concentration at the interface. Figure 3b embodies the angle changes in the carbonyl direction vector relative to the $z$-axis ($\theta_1$) and the dihedral angle ($\theta_2$) among O···S···C···O. Initially (0−75 ps),

the average value of $\theta_2$ (red line) is 65.6°, indicating that the C=O bond orientation is closer to the water slab than to the gas phase. Meanwhile, $\theta_2$ (blue line) varies within a range of 80−160°, suggesting a weak binding between HCHO and HOSO$_2^-$ in the reactant complex. Upon HMS formation, $\theta_2$ stabilizes around ~72°, due to the repulsion interaction between the electronegative oxygen at C−O and S−O, which constrains the rotation of the O−S−C−O dihedral angle. As the simulated time reaching to 77.17 ps, the C−S bond forms an angle of 37.1° along the surface normal (Supplementary Fig. 10), consistent with the vibrational sum frequency spectroscopy calculation (~30°) by Nicholas et al.[54]. This provides solid evidence for the role of interfacial orientations of reactant in modulating heterogeneous reaction. As shown in Fig. 3c, the S···O bond length from 76.70 to 77.15 ps is shortened from

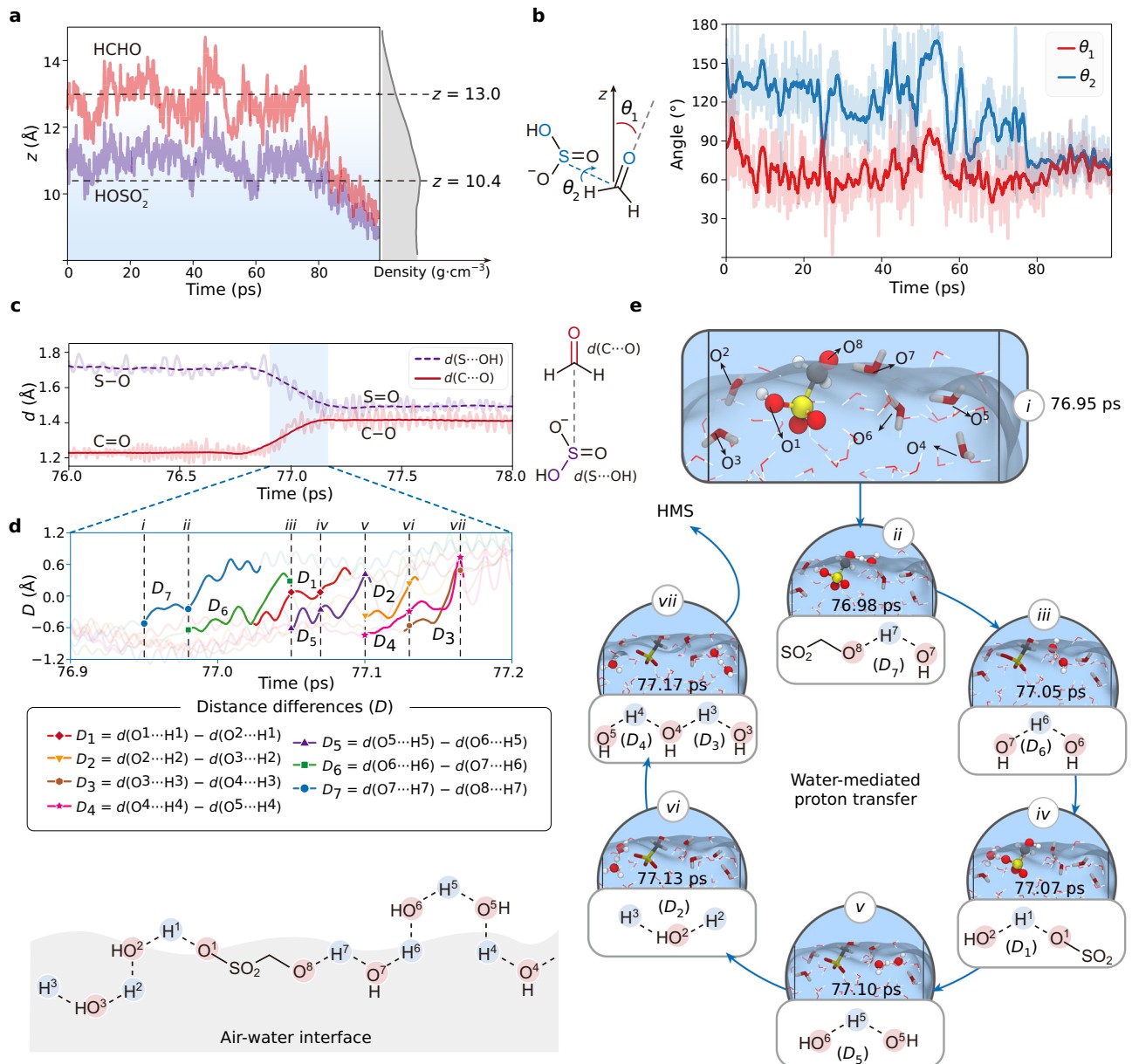

**Fig. 3 | Water-mediated proton transfer at the air-water interface. a** Variation of COM $z$-positions for HCHO (red line) and $HOSO_2^-$ (purple line) as a function of the simulation time with average density along $z$-axis (grey region). Two black dash lines represent the outmost ($z = 13.0$ Å) and the subsurface ($z = 10.4$ Å). **b** Variation of the angle between the C=O direction vector and the $z$-axis (at $\theta_1$, 0–180°, red line) and the dihedral angle between C=O direction vector and the S–OH direction vector (at $\theta_2$, 0–180°, blue line) as a function of the simulation time. Shaded areas are the corresponding angles of every frame. Bold lines are the smoothed value (2000 points) to clarify the trend. **c** Variation of C···O (red line) and S···OH (purple dash line) distances as a function of the simulation time. Shaded areas are the

corresponding distances of every frame. Bold lines are the smoothed value (500 points) to clarify the trend. **d** (Top) Relevant O···H distance differences ($D_1$–$D_7$) variation as a function of the simulation time. The symbols indicate the corresponding $D$ value at the events (*i* to *vii*). (Middle) Definition of the distance differences ($D_1$–$D_7$). (Bottom) Schematic representation of the water-mediated proton transfer at the air-water interface. Red and blue marks represent the O and H atoms participating the proton transfer. Relevant pathway is represented as dash lines. **e** Snapshots of the events (*i* to *vii*) of the water-mediated proton transfer process extracted from the metadynamics-biased AIMD simulations of the heterogeneous reaction of HCHO + $HOSO_2^-$.

-1.7 to -1.5 Å, indicating a transition from the S–O single bond to S=O double bond. At the same period, the C···O bond length is elongated from -1.2 to -1.4 Å, mirroring the C=O double bond changed to C–O single bond.

Water molecules can form hydrogen-bond to surround HCHO and $HOSO_2^-$, facilitating proton transfer through water-mediated hydrogen bond networks. The proton transfer from $HOSO_2^-$ to HCHO through Grotthuss mechanism at the air-water interface is plotted in Fig. 3d and e, and the corresponding analysis for the aqueous solution is presented in Supplementary Fig. 11. Figure 3d delineates the

variation of distance differences ($D_1$ to $D_7$) as a function of simulation time, with the relevant oxygen and hydrogen atoms labeled as $O^i$ ($i = 1$–8) and $H^i$ ($i = 1$–7), respectively. These distance differences are defined as the distance between the donor oxygens and the protons [$d(O^i···H^i)$, $i = 1$–7] and the distance between the protons and the acceptor oxygens [$d(O^{i+1}···H^i)$, $i = 1$–7]. These shifts in $D_1$ to $D_7$ (bold lines) can be regarded as a characteristic of the proton transfer from one water molecule to another. The grey dash lines (labeled as *i*–*vii*) in Fig. 3d correspond to the snapshots of the key events shown in Fig. 3e. Overall, the proton transfer at the air-water interface is

completed in ~0.2 ps. From 76.95 to 77.05 ps (*i* to *iii*), the proton transfer ($D_7$ and $D_6$) takes place in the vicinity of the carbonyl oxygen ($O^8$). Simultaneously, the process of (*iii*) shows that the hydroxyl proton ($H^1$) from $HOSO_2^-$ is transferred to its neighboring water molecule ($D_1$). From 77.05 to 77.10 ps (*iii* to *v*), the proton is transferred to the intermolecular water molecules ($D_5$ and $D_2$), slightly farther from the reaction center. Finally, the proton transfer occurs synchronously at two water molecules ($D_4$ and $D_3$) located at the boundary of the simulation box from 77.10 to 77.17 ps (*v* to *vi*). These findings demonstrate that proton transfer occurs through a dynamic and non-specific network of water molecules, rather than an intramolecular mechanism in gas-phase reactions.

## Accelerated mechanism of HMS formation at the air-water interface

To unveil the accelerated mechanism of the $HCHO + HOSO_2^-$ reaction at the air-water interface, high-level quantum chemical calculations combined with the charge decomposition analysis (CDA)[59,60] were employed. Figure 4a illustrates the relative energy of the reactant complexes, transition states, and products for $HCHO + HOSO_2^- + H_2O$ (red lines, labeled as **R′**, **TS′**, and **P′**) and $HCHO + HOSO_2^- + 4H_2O$ reaction (purple dash lines, labeled as **R″**, **TS″**, and **P″**) in gas phase. Additionally, Supplementary Fig. 18 summarized the high-level quantum chemical calculations for $HCHO + HOSO_2^- + (H_2O)_n$ ($n = 0, 1, 2, 3,$ and 4) reactions. The reaction free-energy barrier for $n = 0, 1, 2, 3,$ and 4 are 14.0 kcal/mol, 11.9 kcal/mol, 11.4 kcal/mol, 10.8 kcal/mol, and 10.5 kcal/mol, respectively. Given previous research discussing the energy barrier reduction associated with water-assisted proton transfer loop formation in the transition state[61–63], we modeled a water molecule as the participator during proton transfer for $n = 1, 2, 3,$ and 4, with the other water molecules acting as spectators that form hydrogen-bond complexes with HCHO and $HOSO_2^-$. Compared to the **TS** without water molecule, the water-mediated proton transfer channel in **TS′** passes through a seven-membered ring (S···O···H···O··· H···O···C), which alleviates the steric strain. As shown in Fig. 4b, the carbonyl electrophilic index in $HCHO-H_2O$ complex (0.35) is slightly higher than that of isolated HCHO molecule (0.32), implying that the water polarization effect enhances the electrophilicity of the carbonyl in HCHO, because the water molecule acts as the Lewis acid to interact the HCHO functioning as the Lewis base[64–66].

Figure 4c provides an overview of the orbital composition of the water-mediated proton transfer by CDA method. The orbital composition of **TS** (black lines) and **TS′** (red dash lines) are summarized in the upper graph in Fig. 4c. For **TS** without water molecule, the formation of the S–C σ orbital is regarded as the overlap between the *n* orbital of $HOSO_2^-$ and the π* orbital of carbonyl. According to the electrophilic index in Fig. 4b, the water molecule is placed with fragment containing HCHO in the transition state for **TS′**. The electronic distribution does not change when water participating the proton transfer. However, the energy level of *n* orbital (highest occupied molecular orbital, HOMO) of the $HOSO_2^-$ in **TS′** is lifted by 0.34 eV in comparison of the $HOSO_2^-$ in **TS**, while the energy level of carbonyl π* orbital (lowest unoccupied molecular orbital, LUMO) in **TS′** is declined by 1.59 eV compared to that in **TS**, suggesting that the water molecule may change the reactivity of the lone-pair-electrons at $HOSO_2^-$ and the π* orbital of the carbonyl.

Furthermore, RxDFT analysis was carried out to quantitatively differentiate the contribution of polarization and solvation (Fig. 4d) in aqueous reaction, compared to gaseous reaction. The polarization contributes to an energy reduction of 2.1 kcal/mol for the transition state (**TS**), but solvation induces an energy increase of 5.0 kcal/mol, falling within the range of 2–20 kcal/mol reported by Wei et al.[67] This increase can be attributed to the solvent barrier effect, which implies that the reactants overcome solvation reorganization as they approach each other during the formation of the transition state[67]. Next, the

influence of the HMS formation in aqueous solution can be attributed to the multifactional influences, including water solvation, water polarization, and activation entropy. Therefore, a quantitative assessment of the influence of the reaction barrier in aqueous solution compared to gaseous reaction can be inferred as Eq. (4).

$$\Delta\Delta G_{b-g}^{\ddagger} = \Delta G_{bulk}^{\ddagger} - \Delta G_{gas}^{\ddagger} = \Delta E_{sol} + \Delta E_{pol} + \Delta E_{pro} \qquad (4)$$

Here, $\Delta\Delta G_{b-g}^{\ddagger}$ represents the difference between the activation Gibbs free-energy in aqueous solution and in gas phase. As shown in Fig. 2b and c, the energy barriers in gas phase ($\Delta G_{gas}^{\ddagger}$) and in aqueous solution ($\Delta G_{bulk}^{\ddagger}$) equal to 14.0 kcal/mol and 11.4 kcal/mol, respectively, resulting in an energy difference of −2.6 kcal/mol. $\Delta E_{sol}$, $\Delta E_{pol}$, and $\Delta E_{pro}$ represent the activation energy contribution by water solvation, water polarization, and water-mediated proton transfer, respectively. According to the RxDFT calculations (Fig. 4d), $\Delta E_{sol}$ and $\Delta E_{pol}$ equal to 5.0 kcal/mol and −2.1 kcal/mol, respectively. Therefore, from Eq. (4), the $\Delta E_{pro}$ value from our theoretical investigations corresponds to the difference in reaction activation entropy between intramolecular proton transfer in gas phase and the water-mediated proton transfer in solution phase, estimated to be −5.5 kcal/mol, well consistent with the experimental result by Boyce et al.[35] ($\Delta E_{pro}^{exp}$ = −5.4 kcal/mol, details provided in Supplementary Note 12).

To date, although various mechanisms have been proposed to elucidate the nature of interfacial acceleration, such as strong electric field (~16 MV/cm)[68–70], charge transfer[71], extreme pH[63], partial solvation[67,72], and molecular orientation at the air-water interface[73], the accelerated mechanism remains unclear and even controversial, due to the lacking quantitative description of the contributions of each factor. For example, Liang et al.[71] demonstrated the predominant influence of the interfacial charge transfer of the interfacial catalysis of reaction involving Criegee intermediate, while Song et al.[69] predicted that the interfacial electric field mainly facilitated the Menshutkin reaction at the air-water interface. The electron density difference of the transition state of the interfacial reaction (Supplementary Note 3 and Supplementary Fig. 12) shows that while the reactants are polarized by water molecules at the air-water interface, no significant charge transfer occurs. To explore the influence of the interfacial electric field, Fig. 4e and f involved the calculation of the energy barriers for the $HCHO + HOSO_2^-$ reaction with and without an external electric field of 0.1 V/Å, respectively. Geometric structures with the electric field were labeled as **R_E**, **TS_E**, and **P_E**, (blue dash lines). The direction of external electric field is opposite to the dipole vector of **TS** (Supplementary Fig. 17). When adding an electric field of 0.1 V/Å, the energy barrier decreases to 12.1 kcal/mol. The energy diminution by electric field ($\Delta E_{ele}$ = 1.9 kcal/mol) originates from the activation of **R_E** relative to **R** and the stabilization of **TS_E** relative to **TS**.

Moreover, wave function analysis is carried out in pursuit of a more comprehensive analysis to the electronic reorganization induced by the electric field. As depicted in Fig. 4e, the region around the carbonyl carbon of HCHO, highlighted in red, indicates a positive electrostatic potential, signifying that the carbonyl carbon serves as the electrophilic site. In contrast, the sulfur atom of $HOSO_2^-$, which exhibits a more negative electric potential, acts as the nucleophilic site. The hydrogen atom, which carries a positive charge at the hydroxyl group of $HOSO_2^-$, is positioned to transfer a proton to the carbonyl oxygen. The ESPs of **TS_E** compared with **TS** illustrate that the external electric field increases the electrostatic negativity of the nucleophilic site and the electrostatic positivity of the electrophilic site. Figure 4f displays the electron density difference between **TS** and **TS_E**, expressed as $\rho(\textbf{TS}) - \rho(\textbf{TS_E})$. The shape of the negative region (blue) at sulfur mirrors the lone-pair-electrons of the sulfur atom and one of the positive regions is distributed over the π* orbital at HCHO. These results suggest that the external electric field alters the ESPs, affecting the electron polarization of the molecular orbitals. This polarization

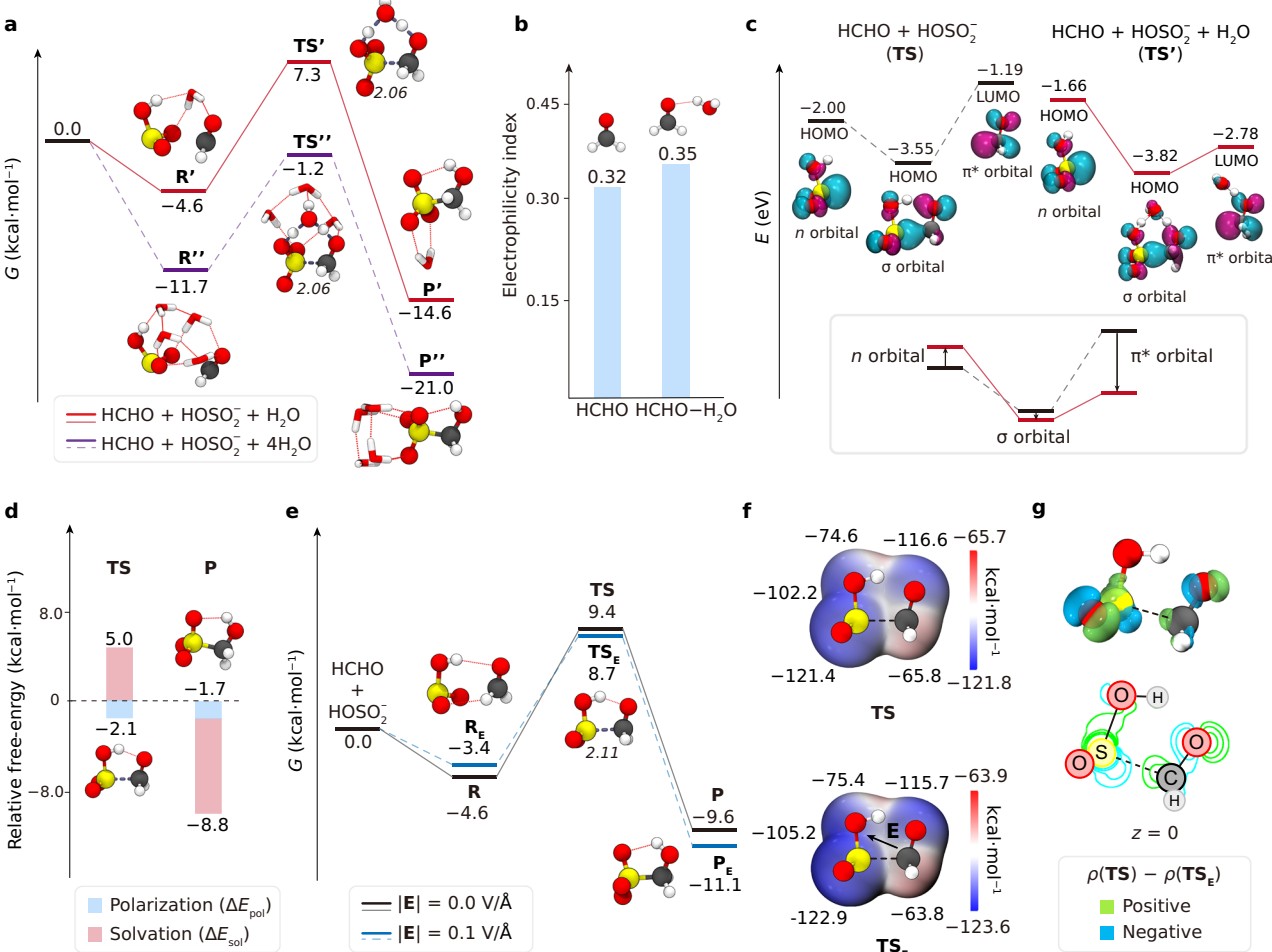

**Fig. 4 | Effect of solvation and interfacial electric field on HCHO + HOSO$_2^-$ reaction. a** Comparison of Gibbs free-energy for the HCHO + HOSO$_2^-$ + H$_2$O (red lines, stationary structures labeled as **R'**, **TS'**, and **P'**) and the HCHO + HOSO$_2^-$ + 4H$_2$O reaction in the gas phase (purple dash lines, stationary structures labeled as **R''**, **TS''**, and **P''**). The calculation level is same as Fig. 2b. **b** Calculated electrophilicity index for the carbonyl carbon for HCHO−H$_2$O complex and HCHO molecule. **c** (Top) CDA analysis for **TS** (black lines) and **TS'** (red lines), relative orbital energies (eV), and the isovalue surfaces of their frontier orbitals (isovalue = ±0.005) of the fragments. (Bottom) Variation of orbital energy with (red) and without (black) a water molecule as proton transfer participant. **d** Relative free-energy contribution of **TS** and **P** for HCHO + HOSO$_2^-$ reaction by

RxDFT calculations, blue: polarization, red: solvation. **e** Comparison of Gibbs free-energy for HCHO + HOSO$_2^-$ reaction with (blue dash lines, stationary structures labeled as **R$_E$**, **TS$_E$**, and **P$_E$**) and without (black lines) an external electric field of 0.1 V/Å in gas phase. The calculation level is same as Fig. 2b. **f** Electrostatic potential surface of **TS** (Top) and **TS$_E$** (Bottom). Red denotes regions of positive electrostatic potential and blue represents regions of negative potential. **g** (Top) The isodensity surface (isodensity = ±0.0015) of the electron density difference between **TS** and **TS$_E$**. Green denotes regions of positive value and blue represents regions of negative value. (Bottom) Contour representation of the electron density difference in the *x-y* plane. Green lines denote positive value and blue lines denote negative value.

plays a key role in the formation of the C−S bond, leading to a lower energy state for **TS$_E$** compared to **TS**. Additional evidence from the LBO analysis validates the electron polarization driven by electric field. The Laplacian bond order of the S−OH bond in **TS$_E$** (0.40) is slightly smaller than that in **TS** (0.41) (Supplementary Fig. 20), suggesting an increased bond polarity of the S−OH bond in the presence of an electric field. Taken together, the effect of an external electric field is attributed to the enhancement of the electrophilic and nucleophilic character of the reactants, facilitating the reaction at the air-water interface.

From a thermodynamic view, the heterogeneous environment brings the impact on molecular translation, rotation, and vibration. Preliminarily, the molecular translation at the air-water interface has been investigated by transferring HCHO or HOSO$_2^-$ molecule from the bulk water across the air-water interface to the gas phase. We employed umbrella sampling techniques to obtain the free-energy profile during these simulations. In Fig. 5a, the Gibbs free-energy for HOSO$_2^-$ (purple dash line) undergoes a dramatic increase after reached

the air-water interface boundary ($z$ = 10 Å). Thus, the subsurface ($z$ = -11 Å, black dash line) tends to be a favorable position to react with HCHO with a slightly increase of ~0.5 kcal/mol in energy. For HCHO (red line), the free-energy at outmost surface ($z$ = -15.6 Å, black dash line) decreases by ~1.1 kcal/mol with the free-energy in bulk water as a reference point, which shows agreement with the QM/MM simulation (1.5 kcal/mol) by Marilia et al.[73]. According to the Boltzmann distribution in Eq. (5), the ratio of probabilities of two states depends on the states' energy difference.

$$\frac{P(i)}{P(j)} \propto e^{\frac{\varepsilon_j - \varepsilon_i}{k_B T}} = e^{\frac{N_A(\varepsilon_j - \varepsilon_i)}{RT}} \qquad (5)$$

where $P(i)$ and $\varepsilon_i$ represent the probability and the molecular energy of the system being in state $i$, respectively, and $N_A$ is the Avogadro constant. Substituting the relative Gibbs free-energy values into Eq. (5) as $N_A\varepsilon_{bulk} = 0$ and $N_A\varepsilon_{inter} = -1.1$ kcal/mol with R$T$ = 0.6 kcal/mol, the ideal state distribution ratio $P$(bulk)/$P$(inter) in the HCHO saturated solution

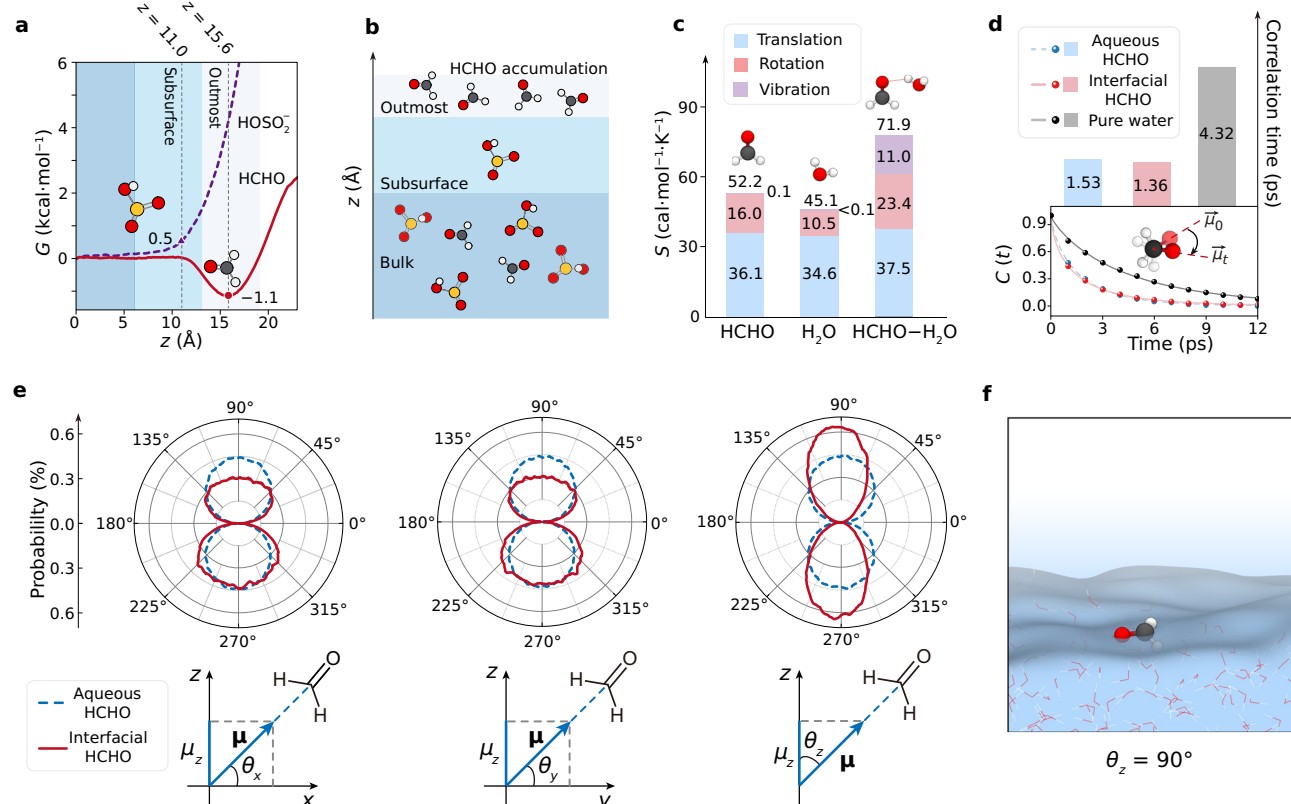

**Fig. 5 | Interfacial accumulation and stabilization. a** Gibbs free-energy profiles for the calculation of a HCHO molecule (red line) or a $HOSO_2^-$ molecule (purple dash line) transferring from the bulk water to gas-phase. **b** Schematic figure of the distribution of the HCHO and the $HOSO_2^-$ molecules in the bulk water and at the air-water interface. **c** Stacked histograms of the calculated entropy of the HCHO molecule, $H_2O$ molecule, and the HCHO−$H_2O$ complex. **d** Dipole autocorrelation function (dots) and the fitted curve (lines) for pure water (grey line), a HCHO molecule in the aqueous system (blue dash line), and a HCHO molecule at the air-water interface (red line). The histograms represent the correlation time for each system. **e** Angular distribution profile (0–360°) for the angle between the C=O direction vector and $x$, $y$, $z$ axis in the bulk water (blue dash lines) and at the air-water interface (red lines). **f** Snapshot of the $\theta_z = 90°$ of HCHO at the air-water interface.

is 6.3. Nevertheless, in the actual saturated solution, the mass transfer of solute is compromised by the alternation of chemical potential ($\mu_i dn_i$) and the surface tension ($\sigma ds$) contributions:

$$dG = -SdT + Vdp + \sum_{i=1}^{N} \mu_i dn_i + \sigma ds \qquad (6)$$

which leads to a lower distribution ratio (see Supplementary Note 10). Our simulation of HCHO saturated solution shows that the surface numerical density of HCHO molecule is ~2.2 times higher than that in the bulk water (Supplementary Fig. 23). By integrating these findings with the AIMD reaction simulations, it is clear that during the heterogeneous formation of HMS, the molecular transition of the two reactants is hindered at the air-water interface, resulting in an overall translational stabilization energy of ~0.6 kcal/mol. Additionally, the accumulation of HCHO at the interface (Fig. 5b) increases the probability of effective collisions, facilitating the reaction at the air-water interface.

The analysis of vibrational degree for HCHO−water system (Fig. 5c) reveals the different contributions of the configurational entropy of the HCHO molecule, $H_2O$ molecule, and HCHO−$H_2O$ complex. For these isolated molecules, the entropy is mainly contributed by translational entropy and the rotational entropy. Upon HCHO−$H_2O$ complex formation, the molecular entropies of complex (71.9 cal·mol⁻¹·K⁻¹) decrease by 25.4 cal·mol⁻¹·K⁻¹ in comparison of the sum between molecular entropies of HCHO (52.2 cal·mol⁻¹·K⁻¹) and $H_2O$ (45.1 cal·mol⁻¹·K⁻¹) molecule. However, the vibrational entropy

increases significantly by ~11.0 cal·mol⁻¹·K⁻¹. This increase is related to the out-of-phase stretching vibration of the carbonyl group, which is the imaginary frequency in the HCHO + $HOSO_2^-$ reaction to form the C−S bond. As shown in Supplementary Fig. 22, the corresponding wavenumbers in the HCHO−$H_2O$ complex (1236.42 cm⁻¹) is larger than that in the isolated HCHO molecule (1218.30 cm⁻¹), which suggests that the $H_2O$ molecule can influence the reactivity of HCHO through vibrational couple.

Next, the investigations of molecular rotation at the air-water interface are summarized in Fig. 5d to f. We initially focused on the dipole autocorrelation function for the HCHO molecule. The definition of the dipole autocorrelation function is given in Eq. (13). As shown in Fig. 5d, the calculated dipole autocorrelation functions are illustrated as solid dots, while the fitted curves are represented as solid lines. The details of the definition and the fitted curve by using assumed Kohlrausch−Williams−Watts stretched exponential[74–76] are shown in Eq. (14) and Supplementary Note 9. The autocorrelation function of pure water (black line) is used as a benchmark. The autocorrelation time of water in our simulation is 4.32 ps, which is consistent with the result of Kumar et al.[77] (4.9 ps in 300 K). The two autocorrelation functions for HCHO in different environment are similar, with the correlation time for the air-water interface system (1.36 ps) slightly shorter than that in the aqueous solution (1.53 ps). Additionally, we analyzed the distribution of the tilt angle between carbonyl in HCHO and the $x$, $y$, $z$ axes represented as $\theta_x$, $\theta_y$, and $\theta_z$, respectively. The definition of $\theta_x$, $\theta_y$, and $\theta_z$ are described in Eq. (12), details provided in Supplementary Note 8. As illustrated in Fig. 5e, three directions of the

angle distribution for HCHO in aqueous solution (blue dash lines) are quite uniform in all directions. However, at the air-water interface, the distribution becomes asymmetric, indicating that the interface constrains the tilt angle. According to the $x$ and $y$ direction, the larger distribution of the tilt angle from 180° to 360° represents the carbonyl oxygen tends to point towards the liquid water to form a hydrogen bond with water but the left-right asymmetry of the $z$-direction distribution. Moreover, the narrower $z$-direction distribution at the air-water interface with a maximum at $\theta_z = 90°$ (Fig. 5f) indicates that the air-water interface can constrain the molecular rotation of HCHO.

The preceding analyses of electric field effects, water polarization, water solvation, and interfacial stabilization provide a quantitative framework for understanding the enhanced reaction rate at the air-water interface compared to the aqueous environment. From a quantitative standpoint, the contribution factors can be elucidated as:

$$\Delta\Delta G_{i-g}^{\ddagger} = \Delta G_{inter}^{\ddagger} - \Delta G_{gas}^{\ddagger} = \Delta E_{psol} + \Delta E_{pol} + \Delta E_{pro} + \Delta E_{ele} + \Delta E_{sta} \quad (7)$$

Here, $\Delta\Delta G_{i-g}^{\ddagger}$ represents the difference between the activation Gibbs free-energy at the air-water interface and in gas phase. According to the results in Fig. 2b and e, the energy barriers in gas phase ($\Delta G_{gas}^{\ddagger}$) and at the air-water interface ($\Delta G_{inter}^{\ddagger}$) equal to 14.0 kcal/mol and 7.6 kcal/mol, respectively, which results in an energy difference of −6.4 kcal/mol. The energy contribution of water polarization and water-mediated proton transfer are approximately equivalent to those in aqueous solution ($\Delta E_{pol} = -2.1$ kcal/mol, $\Delta E_{pro} = -5.5$ kcal/mol). For the additional factor of interfacial stabilization, the interfacial stabilization energy for molecular translation is estimated to be 0.6 kcal/mol, and the rotational stabilization energy can be assessed as approximately 1/2 R$T$, which is around 0.3 kcal/mol at room temperature based on the equipartition theorem. Therefore, the total contribution from interfacial stabilization ($\Delta E_{sta}$) is estimated to be −0.9 kcal/mol. Meanwhile, the energy reduction due to the interfacial electric field ($\Delta E_{ele}$) is calculated to be −1.9 kcal/mol. Furthermore, according to the result in Fig. 5a, the partial solvated HCHO at outmost surface exhibits intermediate-size solvent barriers, smaller than that in the bulk aqueous solution[67]. This means the HCHO needs to overcome a weaker solvation energy to approach $HOSO_2^-$. To distinguish the contribution of water solvation ($\Delta E_{sol}$) introduced in Eq. (4), we define the energy contribution of the partial solvation as $\Delta E_{psol}$. According to Eq. (7), by substituting the calculated values of $\Delta E_{pol}$, $\Delta E_{pro}$, $\Delta E_{sta}$, and $\Delta E_{ele}$, $\Delta E_{psol}$ at the air-water interface can be estimated 4.0 kcal/mol smaller than $\Delta E_{sol}$ (5.0 kcal/mol) for the aqueous solution.

## Heterogeneous HMS formation under weak acidic conditions

The presence of $SO_3^{2-}$ also influences the formation of HMS in weak acidic or neutral conditions[27,34,37]. The reaction mechanism of the $HCHO + SO_3^{2-}$ reaction resembles that of $HCHO + HOSO_2^-$ (see Supplementary Fig. 3), where the $SO_3^{2-}$ functions as the nucleophile, but there is no proton transfer after nucleophilic attack. Figure 6b summarized the snapshot during the metadynamics-biased AIMD simulation for the homogeneous (top) and heterogeneous (bottom) reaction. The calculated free-energy barrier in aqueous phase is 5.8 ± 0.1 kcal/mol (Fig. 6a, blue dash line), which is close to the experimental result[35] ($\Delta G_{exp}^{\ddagger} = 7.1$ kcal/mol) within the relative error margin. This suggests that the reaction proceeds with a moderate barrier in bulk solution. We further employed TI-AIMD simulation to validate this metadynamics result (Supplementary Note 15 and Supplementary Fig. 30). We find that the free-energy barriers obtained from metadynamics and TI-AIMD simulations ($\Delta G_{TI}^{\ddagger} = 6.4 \pm 0.7$ kcal/mol) are consistent within the margin of error, thereby reinforcing the reliability of the metadynamics simulations. As a result, although the metadynamics simulations likely did not reach convergence due to the lack of recrossing events, the resulting free energy profiles and corresponding conclusions have been supported by TI-AIMD. At the

air-water interface, the $HCHO + SO_3^{2-}$ reaction also shows a significant decrease with a free-energy barrier of 2.7 ± 0.5 kcal/mol (Fig. 6a, red line), highlighting the interfacial acceleration effect. The corresponding CV evolution curves between initial state ($\mathbf{R}_{bulk}/\mathbf{R}_{inter}$) and final state ($\mathbf{P}_{bulk}/\mathbf{P}_{inter}$) for each three independent metadynamics simulations are presented in Supplementary Figs. 8 and 9, offering further details on the dynamics and structural evolution during the reaction in both phases.

The high-level quantum chemical calculation of free-energy profiles of the gas-phase reactions $HCHO + SO_3^{2-} + (H_2O)_n$ with $n = 1, 2, 3,$ and 4 was calculated as the same level as the $HCHO + SO_3^{2-}$ reaction. The energy barriers for $HCHO + SO_3^{2-} + (H_2O)_n$ with $n = 0, 1, 2, 3,$ and 4 reactions are 3.4 kcal/mol, 4.4 kcal/mol, 5.1 kcal/mol, 5.9 kcal/mol, and 6.2 kcal/mol, respectively (see Supplementary Fig. 19), due to the strong solvation barrier effect. The reaction energy barrier of $HCHO + SO_3^{2-} + (H_2O)_4$ system is close to the aqueous energy barrier (5.8 kcal/mol) predicted by AIMD simulations. The investigation of the influence by interfacial electric field and interface stabilization is similar to the result of $HCHO + HSO_3^-$ reaction. As shown in Fig. 6c, the free-energy barrier of $G(\mathbf{TS_E}) - G(\mathbf{R_E})$ was found to be 2.6 kcal/mol lower than $G(\mathbf{TS}) - G(\mathbf{R})$. The electrostatic potential surface (Fig. 6d) indicates that the electric field increases the electronegativity of the nucleophilic $SO_3^{2-}$ and the electron positivity of the electrophilic HCHO. According to the electron density difference (Fig. 6e), upon the addition of electric field, the electron density in $n$ orbital of $SO_3^{2-}$ increases, but the electron density at the carbonyl $\pi^*$ orbital decreases. Furthermore, molecular dynamics simulation combining umbrella sampling technique was carried out to calculate the Gibbs free-energy variation when moving a $SO_3^{2-}$ from the bulk water to the gas phase (see Supplementary Fig. 25). At the subsurface ($z = -11$ Å), the free-energy of $SO_3^{2-}$ increases by 0.7 kcal/mol compared to the state of $SO_3^{2-}$ in the bulk water, which shows agreement with the trend of numerical density of the simulation of 0.1 M $HOSO_2^-/SO_3^{2-}$ solution by Buttersack et al.[42], where the ratio of numerical density at subsurface between $HOSO_2^-$ and $SO_3^{2-}$ is -1.5.

## Quantitative mechanism of the heterogeneous HMS formation

By consolidating the thermodynamics and the chemical kinetics from $HCHO + HOSO_2^-$ and $HCHO + SO_3^{2-}$ reaction to form HMS, a coherent quantitative mechanism of the HMS formation in atmospheric aerosol can be articulated. As depicted in the left panel of Fig. 7a, the heterogeneous environment results in the accumulation of HCHO at outmost surface. In moderate acidic aerosol (pH = 2–4), the HMS is formed by the nucleophilic addition by HCHO and $HOSO_2^-$. Although the tautomeric equilibrium[42] of $HOSO_2^- \rightleftharpoons HSO_3^-$ ($K_T = 3.2$ at pH = 4) leads to a relative low concentration of $HOSO_2^-$ at the air-water interface. According to the reactivity of $HSO_3^-$ (Supplementary Note 13 and Supplementary Fig. 27), the free-energy barrier ($\Delta G_{sul}^{\ddagger}$) of the $HCHO + HSO_3^-$ reaction at the air-water interface is 18.1 ± 1.0 kcal/mol, which is significantly higher than that of $HOSO_2^- + HCHO$ reaction ($\Delta G^{\ddagger} = -7.6$ kcal/mol). As a result, compared to $HSO_3^-$, $HOSO_2^-$ exhibits substantially higher reactivity toward the surface-accumulated HCHO, thereby accelerating the formation of HMS. Moreover, the tautomeric equilibrium between $HSO_3^-$ and $HOSO_2^-$ enables a continuous supply of the more reactive $HOSO_2^-$ species, sustaining the overall conversion process. This dynamic equilibrium underpins a cascade reaction mechanism, ensuring efficient HMS production under moderate acidic environment. In the right panel, the reaction between the minor amount of $SO_3^{2-}$ and HCHO at the air-water interface and in aqueous solution are complementary pathways with a lower energy barrier of -2.7 kcal/mol and -5.8 kcal/mol, respectively. Due to the p$K_a$ of HMS ($HOCH_2SO_3^-$) is -12[44], the product of $HCHO + SO_3^{2-}$ reaction, $O^-CH_2SO_3^-$, can react with $HOSO_2^-$ to form HMS and regenerate $SO_3^{2-}$ through the $O^-CH_2SO_3^- + HOSO_2^- \rightarrow HOCH_2SO_3^- + SO_3^{2-}$ reaction during current pH conditions.

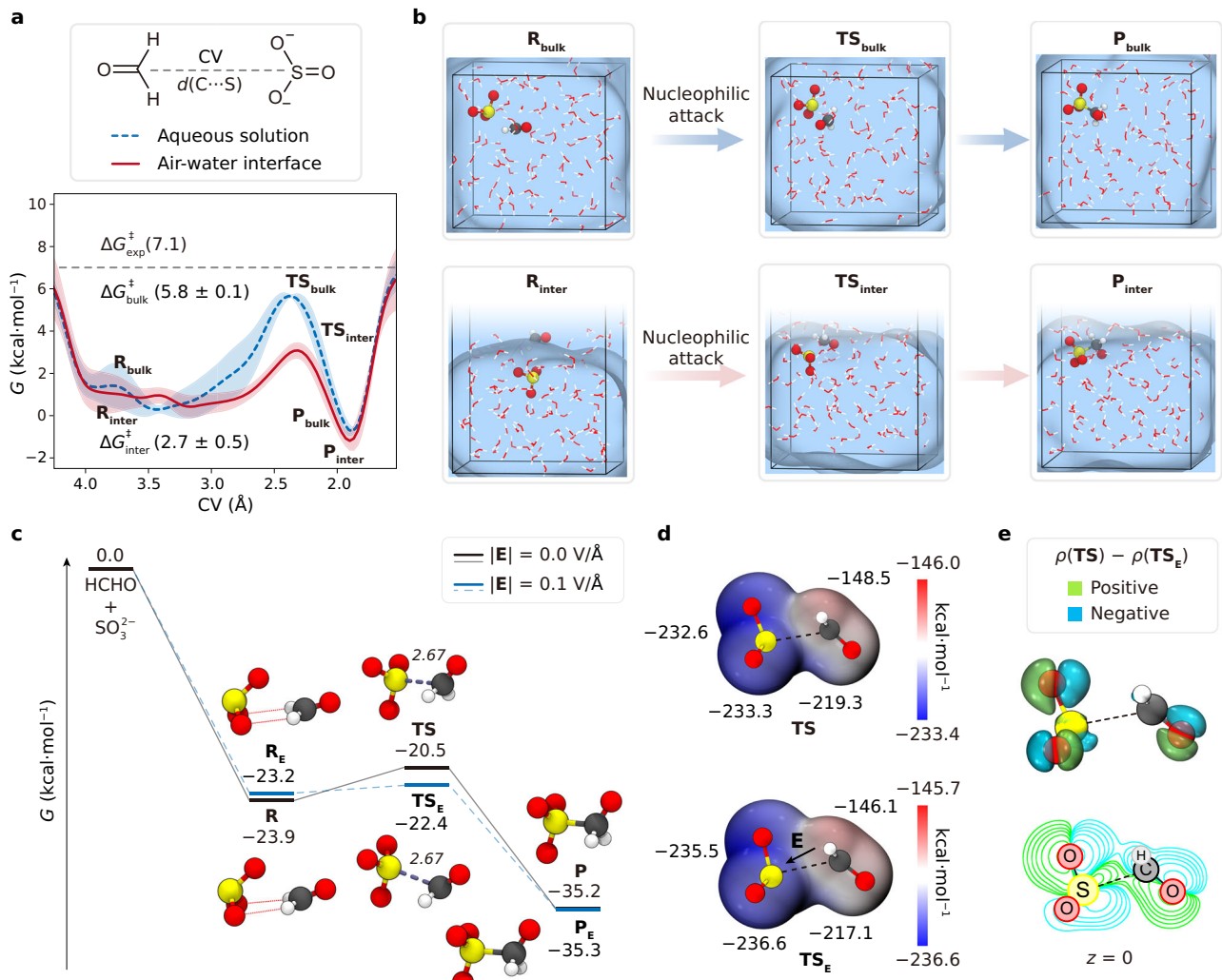

**Fig. 6 | HMS formation at weak acidic or neutral conditions. a** (Top) Schematic representation of the CV. (Bottom) Gibbs free-energy profiles for the HCHO + SO$_3^{2-}$ reaction in bulk water (blue dash line) and at the air-water interface (red line) compared with the experimental result ($\Delta G_{exp}^{\ddagger}$) by Boyce et al.[35]. Shaded areas represent the error bands for each three simulations. Calculation of error band is as same as the HCHO + HOSO$_2^-$ reaction simulation. **b** Snapshot structures obtained from the metadynamics-biased AIMD simulations for the aqueous (Top: **R**$_{bulk}$, **TS**$_{bulk}$, and **P**$_{bulk}$) and the heterogeneous reaction (Bottom: **R**$_{inter}$, **TS**$_{inter}$, and **P**$_{inter}$). **c** Gibbs free-energy profiles for HCHO + SO$_3^{2-}$ reaction with (blue dash lines) and without (black lines) an external electric field of 0.1 V/Å in gas phase and their corresponding structures of the stationary points. The calculation level is same as Fig. 4e. **d** Electrostatic potential surface of **TS** (Top) and **TS**$_E$ (Bottom). Red denotes regions of positive electrostatic potential and blue represents regions of negative potential. **e** (Top) The isodensity surface (isodensity = ±0.0015) of the electron density difference between **TS** and **TS**$_E$. Green denotes regions of positive value and blue represents regions of negative value. (Bottom) Contour representation of the electron density difference in the x-y plane. Green lines denote positive value and blue lines denote negative value.

Figure 7b presents the quantitative mechanism underlying HMS formation through the major pathway of HCHO + HOSO$_2^-$ reaction. Due to the extra activation entropy induced by the intramolecular proton-transfer, the gaseous reaction exhibits a large energy barrier of $\Delta G_{gas}^{\ddagger}$ = 14.0 kcal/mol. However, in the aqueous solution, although the solvent barrier effect increases the activation energy of $\Delta E_{sol}$ = 5.0 kcal/mol, the water molecules can not only form hydrogen-bond with the reactant to provide polarization effect ($\Delta E_{pol}$ = −2.1 kcal/mol) but also bridge the loop of proton transfer channel through Grotthuss mechanism to reduce activation entropy by ~5.5 kcal/mol, which then reduce the reaction energy barrier to $\Delta G_{bulk}^{\ddagger}$ = ~11.4 kcal/mol. At the air-water interface, the partial solvated HCHO reduces the activation enthalpy by ~1.0 kcal/mol, and the interfacial stabilization effect from the frustrated translation and rotation of HCHO further decreases the configurational entropy barrier by ~0.9 kcal/mol ($\Delta E_{sta}$), further complemented by interfacial electric field ($\Delta E_{ele}$ = 1.9 kcal/mol). The cumulative influence of these factors leads to approximately two orders of magnitude enhancement of interfacial reaction rate compared to the aqueous reaction.

Additionally, at extreme acidic condition (pH = 0.8–1.8, Supplementary Note 14 and Supplementary Fig. 28), Buttersack et al.[42] reported that HSO$_3^-$ becomes the predominant species at the air-water interface. The surface-accumulated HCHO can undergo nucleophilic addition with this species to form HMS isomer. As the pH increases to weak acidic or neutral conditions (pH > 4), the concentration of SO$_3^{2-}$ rises significantly. Under such condition, the formation of HMS is energetically favorable, due to a low energy barrier of SO$_3^{2-}$ + HCHO reaction both at the air-water interface and in the aqueous solution (Supplementary Note 14 and Supplementary Fig. 29). Furthermore, the effect of inorganic ions on the reaction energy profile in the bulk solution is investigated. The free-energy barrier for the HOSO$_2^-$ + HCHO reaction in the salt solution ($\Delta G_{ion}^{\ddagger}$) is determined to be 10.2 kcal/mol (Supplementary Note 12 and Supplementary Fig. 26). Compared to the pure bulk solution ($\Delta G_{bulk}^{\ddagger}$ = 11.4 kcal/mol), the presence of salt ion slightly lowers the free-energy barrier but does not

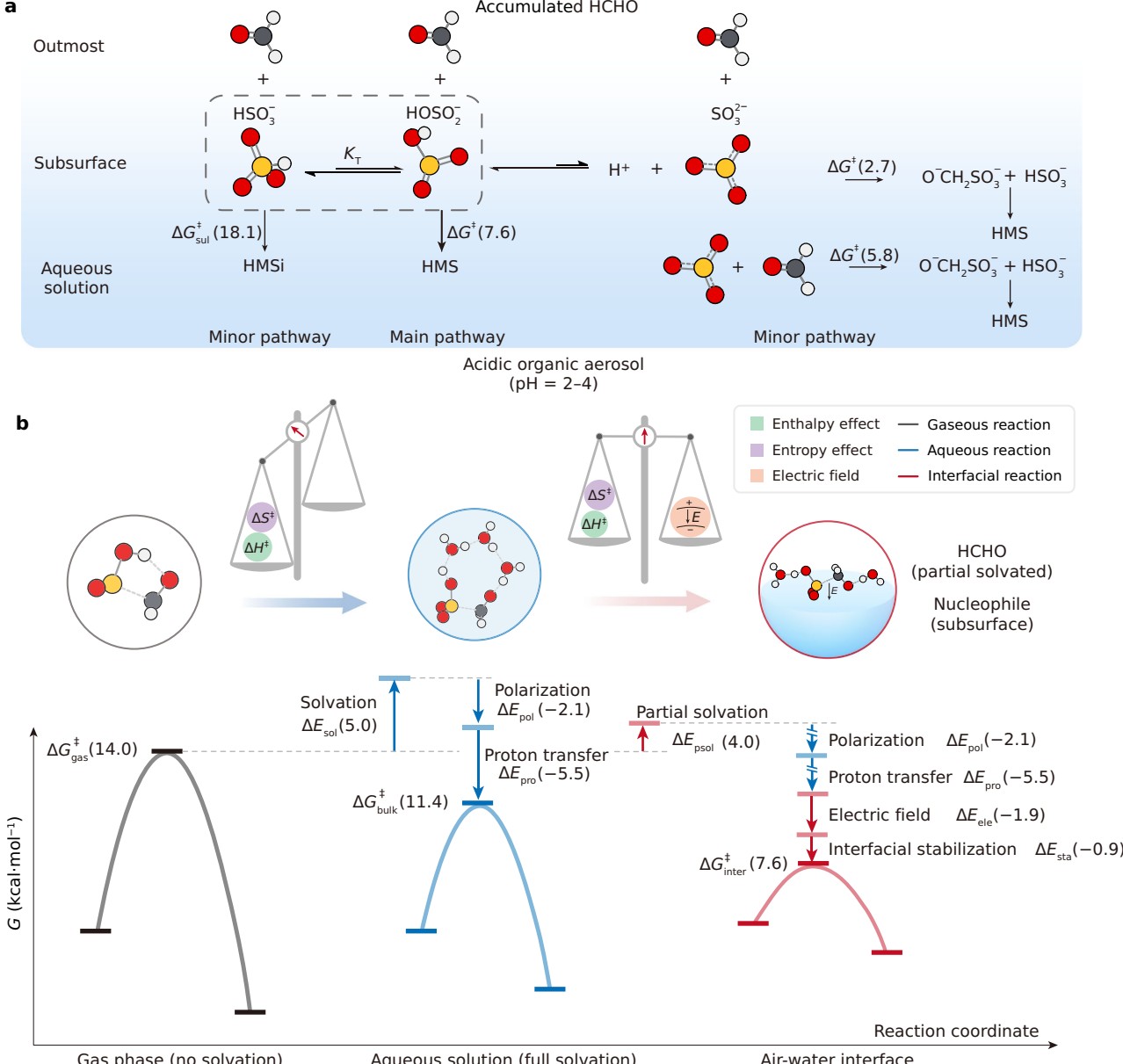

**Fig. 7 | Coherent quantitative mechanism of the HMS formation in atmospheric aerosol. a** The reaction pathways of HMS formation in acidic aerosols. Although $HSO_3^-$ predominates in the equilibrium with $HOSO_2^-$ under moderate acidic conditions, the reaction between $HOSO_2^-$ and surface-accumulated HCHO at the gas-liquid interface proceeds with a relatively low energy barrier ($\Delta G_{inter}^\ddagger$ = -7.6 kcal/mol) compared to the HMS isomer formation ($\Delta G_{sul}^\ddagger$ = -18.1 kcal/mol), leading to the cascade formation of HMS. This process depletes the concentration of $HOSO_2^-$, driving the ongoing conversion of $HSO_3^-$ to $HOSO_2^-$. As a result, $HOSO_2^-$ continuously reacts with HCHO to generate HMS. **b** (Left) The nucleophilic addition with intramolecular proton-transfer mechanism and the corresponding free-energy profile of gaseous HMS formation. (Middle) Free-energy profile for the homogeneous HMS formation with full solvated and solvent polarized HCHO and $HOSO_2^-$ via water-mediated proton transfer process. (Right) Free-energy profile for the heterogeneous HMS formation. The HCHO is partial solvated and the $HOSO_2^-$ is located at subsurface with interfacial electric field and interfacial stabilization.

alter the underlying reaction mechanism. These findings provide a coherent quantitative framework for understanding HMS formation in atmospheric aerosols and highlight the intricate interplay between thermodynamics and chemical kinetics at the air-water interface.

## Discussion

In this study, we combine metadynamics-biased AIMD simulation with high-level quantum chemical calculations to elucidate the heterogeneous formation of HMS, particularly as it pertains to the abundant presence in acidic conditions during extreme cold. Although the role of atmospheric pH in HMS formation is well established, the direct probing of the underlying mechanism at the air-water interface is still

limited. Our findings reveal that, although bisulfite ($HOSO_2^-$) exhibits a higher activation energy for reaction with formaldehyde (HCHO) in the gas phase or aqueous solution, the synergistic regulation of entropy and enthalpy governs the rapid formation of HMS through nucleophilic addition at the air-water interface. This process cannot be solely attributed to the interfacial electric field or partial solvation. According to the RxDFT results, compared to the gas-phase reaction, solvated reactants in the bulk solution must overcome a water reorganization barrier of ~5.0 kcal/mol, partially compensated by a polarization effect of ~2.1 kcal/mol. Notably, the cooperative hydrogen-bonding networks of water molecules facilitate proton transfer via the Grotthuss mechanism, reducing activation entropy by ~5.5 kcal/mol. This lowers

the free-energy barrier in the bulk solution to -11.4 kcal/mol, closely agreeing with experimental data. At the air-water interface, the partial solvation of HCHO further decreases the activation enthalpy by -1.0 kcal/mol. Additionally, the interfacial stabilization arising from restricted translation and rotation of HCHO reduces the configurational entropy barrier by -0.9 kcal/mol. This synergistic regulation of entropy and enthalpy effect is comparable to interfacial electric field effect with a reduction of 1.9 kcal/mol, leading to a remarkable two orders of magnitude enhancement of interfacial reaction rate compared to the aqueous reaction. This study provides a mechanistic framework for understanding the formation of HMS in acidic atmospheric environments and offers insights that could provide strategies for air quality management, particularly in regions susceptible to winter haze episodes. Furthermore, as a prototypical example of nucleophilic addition to carbonyl compounds, the quantitatively decoding the multifactorial influences on the accelerated reaction kinetics will tap the robust catalytic potential of microdroplets, paving the way for the future development of green and efficient synthetic strategies.

## Methods

In this work, we used metadynamics-biased ab initio molecular dynamics (AIMD) simulations, high-level quantum chemical calculations, classic molecular dynamics simulations, and RxDFT calculations to elucidate the nucleophilic addition mechanism between HCHO and $HOSO_2^-/SO_3^{2-}$. The overall workflow is summarized in Supplementary Fig. 1.

### Metadynamics-biased AIMD simulations

Metadynamics-biased AIMD simulations[78] were employed to elucidate the reaction dynamics between HCHO and $HOSO_2^-/SO_3^{2-}$ using the CP2K[79] package. The simulations incorporated the Becke–Lee–Yang–Parr (BLYP) functional[80,81], complemented by Grimme's dispersion corrections[82] and Becke–Johnson[83] damping term, in conjunction with the double-ζ plus polarization (DZVP) basis set. An energy cutoff of 300 Ry was set for the plane-wave basis set, and core electrons were modeled using Goedecker–Teter–Hutter (GTH) norm-conserving pseudopotentials[84]. To simulate the heterogeneous system, a water slab with two air-water interfaces was constructed, comprising 96 water molecules in a simulation box of $1.5 \times 1.5 \times 3.0$ ($x \times y \times z$) nm³. For bulk water system, a cubic simulation box filled with water molecules, measuring $1.42 \times 1.42 \times 1.42$ ($x \times y \times z$) nm³, was utilized. To ascertain the computational accuracy, an additional simulation of the larger heterogeneous system was performed by using a water slab consisting of 192 water molecules in a simulation box of $1.5 \times 1.5 \times 6.0$ ($x \times y \times z$) nm³. The simulation system of $HOSO_2^- + HCHO$ reaction in salt solution contains one $Na^+$ ion, one $Cl^-$ ion, and 93 water molecules in a cubic periodic boundary box of $1.42 \times 1.42 \times 1.42$ ($x \times y \times z$) nm³. All AIMD simulations were performed in the canonical (NVT) ensemble, with a temperature of 298.15 K maintained via the velocity rescaling thermostat[85] (CSVR) method. The time step for the simulation was set to 1 fs. Relevant error bands of the free-energy profiles are calculated on the standard deviation by employing cubic interpolation on three independently obtained free-energy profiles. The error of the free-energy barrier is the standard deviation of the free-energy barriers of three metadynamics simulations. Details of the metadynamics-biased AIMD simulations are summarized in Supplementary Note 1.

### Thermodynamic integration AIMD simulations

Thermodynamic integration (TI)-AIMD simulation were carried out to investigate the reaction of $HCHO + HSO_3^-$ at the air-water interface and validate the metadynamics results for the $SO_3^{2-} + HCHO$ reaction in the bulk solution. For all simulations, a total of 11 sampling windows were implemented to guarantee the smoothness of the calculated free-

energy profile. Each window was equilibrated for ~15 ps, followed by a production run of 5 ps for the free-energy sampling. All parameters of simulations were performed as same as the metadynamics-biased AIMD simulations. The free-energy values for each window are obtained from three production runs. Relevant error bars of each window are calculated using the standard deviation of the corresponding free-energy values of three production runs. The free-energy profile is generated by piecewise cubic hermite interpolating polynomial (PCHIP) interpolation of calculated free-energy values of three production for each window. Details of the TI-AIMD simulations are provided in Supplementary Note 15.

### High level quantum chemical calculations

The Gibbs free-energy profiles of the gas-phase reactions with and without an external electric field were derived from the high-level quantum chemical calculations as implemented in Gaussian 16[86] software. All stationary geometries were optimized with the M06-2x functional[87] and the 6-311G++(d, p) (for anions)/6-311G(d, p) (for neutral molecules) basis sets[88–90]. The single point energies of all stationary geometries were further obtained at the CCSD(T)/aug-cc-pVTZ level[91,92]. The reactions involving $HCHO + HOSO_2^-$ and $HCHO + SO_3^{2-}$ were calculated under the effect of an external electric field of 0.1 V/Å, while the reactions for $HCHO + HOSO_2^- + (H_2O)_n$ with $n = 1, 2, 3, 4$; reactions with $HCHO + SO_3^{2-} + (H_2O)_n$ with $n = 1, 2, 3$, and 4 were not subjected to the electric field. Details of the high-level quantum chemical calculations and wavefunction analysis are summarized in Supplementary Note 3. The global minimum of the cluster structures was initially searched using ABCluster 3.0 program[93,94] before geometrical optimization. Furthermore, the isosurface graphs of electrostatic potential[95,96] (ESP), the difference of the electron density, the Laplacian bond order[97] (LBO), the charge decomposition analysis[59,60] (CDA), and the electrophilicity index quantified by conceptual DFT[98,99] were calculated by using Multiwfn[100,101] package.

### Classical molecular dynamics simulations

CMD simulations were performed using the GROMACS package[102] with periodic boundary conditions in the NVT ensemble at 298.15 K, controlled using CSVR thermostat. Optimized potentials for liquid simulations all-atom[103] (OPLS-AA) were applied for the reactant molecules (HCHO/$HOSO_2^-/SO_3^{2-}$) and the counterion $K^+$, and the SPC/E model[104] was used for the water molecules. Restrained electrostatic potential[105] (RESP) charge was applied to calculate the atomic charge by using Multiwfn program[100,101]. Nonbonding interactions were described by the Lennard-Jones and Coulomb potentials. The particle-mesh Ewald summation method[106] was employed for the calculation of electrostatic interactions, and a real-space cutoff of 1.0 nm was applied to nonbonded interactions. The LINCS algorithm[107] was used to constrain all hydrogen-containing bonds. The HCHO@$(H_2O)_{1000}$ system was simulated in the bulk water or at the air-water interface, and the HCHO saturated solution. The Gibbs free-energy profile for the transfer of a reactant molecule (HCHO/$HOSO_2^-/SO_3^{2-}$) from the bulk water (1000 water molecules) across the air-water interface to the gas-phase was calculated by using the umbrella sampling (US) method. All GROMACS topology files were generated by Sobtop program[108]. Details of CMD simulations are summarized in Supplementary Note 7.

### RxDFT calculations

Within the theoretical framework of RxDFT[56], the reaction free-energy in solution, $\Delta G_{bulk}^{\ddagger}$, can be expressed as:

$$\Delta G_{bulk}^{\ddagger} = \Delta G_{gas}^{\ddagger} + \Delta E_{sol} \qquad (8)$$

where $\Delta G_{gas}^{\ddagger}$ denotes the intrinsic reaction free-energy in gas phase. $\Delta E_{sol}$ denotes the difference between the solvation free-energy of

product and reactant. $\Delta G_{gas}^{\ddagger}$ can be obtained from the above high-level quantum chemical calculations. $\Delta E_{sol}$ can be computed by using molecular DFT. Molecular DFT (MDFT) calculations are carried out in the grand canonical ensemble with $T = 298.15\,K$ and a density of water is 0.033 particles/$\text{Å}^3$. The structure and charge of a single solute molecule serves as the external potential input to the MDFT calculation. Specifically, we fix the solute molecule at the center of a cubic box, and the interaction between the solute and surrounding water molecules at the position $\mathbf{r}$ with spatial orientation $\Omega$ can be computed. This interaction virtually provides the external potential, $V_{ext}(\mathbf{r}, \Omega)$, to the surrounding water molecules[109],

$$V_{ext}(\mathbf{r}, \Omega) = \sum_i^a \sum_j^b 4\varepsilon_{ij} \left[ \left( \frac{\sigma_{ij}}{r_{ij}} \right)^{12} - \left( \frac{\sigma_{ij}}{r_{ij}} \right)^6 \right] + \frac{q_i q_j}{4\pi\varepsilon_0 r_{ij}} \quad (9)$$

where $i$ and $j$ are the atoms (sites) in solute and water molecule, respectively, and $r_{ij}$ is the interatomic separation. $q_i$ and $q_j$ are the partial charges carried by both atoms. For the water molecule, we adopt the SPC/E model[104], while the OPLS-AA force field are employed to give the LJ parameters of the molecules. The conventional Lorentz–Berthelot combination rule is used to generate the LJ parameters for the crossed pairs, namely $\varepsilon_{ij} = \sqrt{\varepsilon_i \varepsilon_j}$ and $\sigma_{ij} = (\sigma_i + \sigma_j)/2$. The charge distributions on molecules are calculated by fitting the individual (atomic) charges, centered at atomic positions, to the molecular ESP, which is computed from the quantum mechanical wave function. The MDFT calculation is performed with the homemade code[109,110] in a cubic box of size $L = 100\,\text{Å}$.

### Electron density difference at the air-water interface
The definition of the electron density difference $\Delta\rho$ is written by:

$$\Delta\rho = \rho(\mathbf{TS}) - [\rho(HCHO\cdots HOSO_2^-) + \rho(slab)] \quad (10)$$

where $\rho(\mathbf{TS})$ represents the electron density of the transition state (Fig. 2e, $\mathbf{TS}_{inter}$), $\rho(HCHO\cdots HOSO_2^-)$ represents the electron density of the HCHO and $HOSO_2^-$ molecules, and $\rho(slab)$ represents the electron density of all of the water molecules of the TS structure. The calculation are carried out by using Multiwfn program[100,101], results are provided in Supplementary Note 3.

### Interaction free-energy
The definition of interaction free-energy ($\Delta G_{inter}$) is defined by the difference of the Gibbs free-energy:

$$\Delta G_{inter} = G_{HMS-M} - (G_{HMS} + G_M) \quad (11)$$

where M represents the molecule ($H_2O/SO_2/HOSO_2^-$). All Gibbs free-energy are calculated in M06-2x/6-311G++(d, p) level. Results of the interaction free-energy are summarized in the Supplementary Note 6 and Supplementary Table 4.

### The angle between the dipole vector and the $x$, $y$ and $z$ axis
The angles (represented as $\theta_x$, $\theta_y$, and $\theta_z$) between the carbonyl dipole vector $\boldsymbol{\mu}$ ($\mu_x$, $\mu_y$, $\mu_z$) of HCHO and the $x$, $y$ and $z$ axis are defined as [0°, 360°), given by:

$$\theta_i = \begin{cases} \frac{180°}{\pi} \arccos\left( \frac{\boldsymbol{\mu} \cdot \mathbf{e}_i}{|\boldsymbol{\mu}|} \right) & (\mu_z > 0) \\ -\frac{180°}{\pi} \arccos\left( \frac{\boldsymbol{\mu} \cdot \mathbf{e}_i}{|\boldsymbol{\mu}|} \right) + 360° & (\mu_z \leq 0) \end{cases} \quad (12)$$

where $\mathbf{e}_i$ represent the unit vector of the cartesian axis ($i = x$, $y$ and $z$). $|\boldsymbol{\mu}|$ represents the modulus of the dipole vector $\boldsymbol{\mu}$. Details of the calculation are summarized in Supplementary Note 8.

### Dipole autocorrelation function
The definition of the dipole autocorrelation function is given by:

$$C(t) = \frac{1}{N} \sum_{i=1}^{N} \frac{\langle \boldsymbol{\mu}_i(t) \cdot \boldsymbol{\mu}_i(0) \rangle}{\langle \boldsymbol{\mu}_i(0) \cdot \boldsymbol{\mu}_i(0) \rangle} \quad (13)$$

where $\boldsymbol{\mu}_i(t)$ represents the time evolution of the normalized dipole vectors of molecule, $N$ is the numbers of the molecules, and $t$ is the simulation time. To gain the correlation time for each system, we used the assumed Kohlrausch-Williams-Watts stretched exponential for the long-time relaxation behavior of autocorrelation functions $\phi(t)$, as written by mode coupling theory (MCT)[74–77]:

$$\varphi(t) = A e^{-\left( \frac{t}{\tau_a} \right)^\beta} \quad (14)$$

where $\tau_a$ is the correlation time, $\beta$ is the exponent, and $A$ is the fitting parameters. Details of the dipole autocorrelation function calculation are provided in Supplementary Note 9.

### Activation entropy of the proton transfer
The experimental value of $\Delta E_{pro}$ (represented as $\Delta E_{pro}^{exp}$) can be estimated by the reaction activation entropy data by Boyce et al.[35]. Considering the transition state (**TS**) of $HCHO + HOSO_2^-$ (Fig. 2d, $\mathbf{TS}_{bulk}$) and $HCHO + SO_3^{2-}$ (Fig. 6b, $\mathbf{TS}_{bulk}$) reactions in bulk solution from our AIMD simulations. Beyond the common properties of nucleophilic attack from S to C observed in both two **TS**s, the **TS** of $HCHO + HOSO_2^-$ reaction also exhibits distinctive proton-roaming feature. Owing to the fact that the nucleophilic attack and the proton roaming process are nearly independent. The reaction activation entropy for the $HCHO + HOSO_2^-$ reaction ($\Delta S_1^{\ddagger}$) can be estimated as:

$$\Delta S_1^{\ddagger} \approx \Delta S_{nuc}^{\ddagger} + \Delta S_{pro}^{\ddagger} \quad (15)$$

where $\Delta S_{nuc}^{\ddagger}$ represents the contribution of nucleophilic addition and $\Delta S_{pro}^{\ddagger}$ represents the contribution of proton-roaming. Meanwhile, the $\Delta S_{nuc}^{\ddagger}$ approximately equals to the reaction activation entropy for the $HCHO + SO_3^{2-}$ reaction ($\Delta S_2^{\ddagger}$):

$$\Delta S_2^{\ddagger} \approx \Delta S_{nuc}^{\ddagger} \quad (16)$$

Therefore, the equation can be inferred as:

$$\Delta E_{pro}^{exp} \approx T(\Delta S_1^{\ddagger} - \Delta S_2^{\ddagger}) \quad (17)$$

Details of activation entropy calculations are summarized in Supplementary Note 11.

### Influence of salt ions
At low ionic concentrations, Zhang et al.[47] reported a quantitative relationship between the reaction rate constant of the HMS formation and the ionic strength in bulk solution:

$$\lg k = \frac{17.59 I}{8.52 + I} + 0.57 \quad (18)$$

where $I$ represents the ionic strength and $k$ denotes the reaction rate constant. The ionic strength is calculated by the following formula:

$$I = \frac{1}{2} \sum c_i z_i^2 \quad (19)$$

in which $c_i$ is the molar concentration (mol/L) of the ion $i$ and $z_i$ is its charge. Details of calculation of the influence of salt ions are summarized in Supplementary Note 12.

## Data availability

The authors declare that the full computational details and results supporting the findings of this study are available within the paper and its supplementary information files. The computed data[111] including the cartesian coordinates of the molecules and the snapshot extracted from the AIMD/CMD trajectories in the main text and analysis data for drawing figures generated in this study are provided at https://doi.org/10.5281/zenodo.15253851.

## Code availability

The code of RxDFT used within the article is available from the corresponding author upon request.

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

## Acknowledgements

We thank Shendong Tan for useful discussion. J.Y. was supported by National Natural Science Foundation of China (22203032). X.H. was supported by the National Natural Science Foundation of China (Grant Nos. 92477103 and 22273023), Shanghai Municipal Natural Science Foundation (Grant No. 23ZR1418200), the Natural Science Foundation of Chongqing, China (Grant No. CSTB2023NSCQ-MSX0616), the Shanghai Frontiers Science Center of Molecule Intelligent Syntheses, Shanghai Future Discipline Program (Quantum Science and Technology), Guizhou Provincial Science and Technology Projects, China (CXTD 2022001), Shanghai Municipal Education Commission's "Artificial Intelligence-Driven Research Paradigm Reform and Discipline Advancement Program", and the Fundamental Research Funds for the Central Universities. W.T. acknowledges for the financial support from the National Natural Science Foundation of China (Grant No. 22108070) and the Young Elite Scientists Sponsorship Program by CAST (No. 2022QNRC001). We also acknowledge the Supercomputer Center of East China Normal University (ECNU Multifunctional Platform for Innovation 001) for providing computer resources.

## Author contributions

J.Y. and X.H. provided continuous supervision throughout the project. J.Y. and J.L. conceived the ideas and designed the research. J.Y. and J.L. wrote the manuscript, and J.L. carried out all the ab initio molecular dynamics simulations, high-level quantum chemical calculations, and classical molecular dynamics simulations. W.T. carried out the reaction

density functional calculations. J.L., W.T., and J.Z. contributed to data analysis and manuscript preparation.

## Competing interests

The authors declare no competing interests.
