## [Transparent Peer Review file · Nature Communications]

Hydroxymethanesulfonate Formation accelerated at the Air-water Interface by Synergistic Enthalpy-Entropy Effects

Corresponding Author: Professor Xiao He

Version 0:

Reviewer comments:

Reviewer #1

(Remarks to the Author)

In this study, the authors have applied metadynamics-biased ab initio molecular dynamics (AIMD) simulations with high-level quantum chemistry calculations and reaction density functional theory to investigate the reaction mechanisms of HMS formation at the interface. This study does provide a new perspective on the role of heterogeneous reactions on the HMS formation, advancing our understanding of atmospheric chemistry.

I did appreciate the authors have carried out very thorough and detailed model simulations. Overall, the presentation of the results are clear and the discussion concise and thoughtful. The proposed mechanisms are supported by the simulations and arguments are reasonable. In particular, the authors demonstrate the potential role of water molecules at the interface in the reactions and formation of HMS. This is a very nice chemistry and modeling paper. Indeed, I don't have specific comment on the data and discussion (the manuscript can be published nearly in current form).

I did appreciate the authors have carried out very thorough and insightful model simulations. However, my main question (or concern) is that the authors have only considered relative simple systems. How their results and findings could be applied for more complex aerosol mixtures and cloud droplets. For instance, in addition to aerosol acidity or pH, how the presence of other species (e.g. inorganic ions and organic compounds (e.g. organic surfactants)) affect the reaction rates or mechanisms at the interface or bulk solution phase? How the composition and concentration of the species would alter the reaction rates and mechanisms? I thought this would be important for the authors to address or comment such that the findings of this work could have boarder and greater applications.

Reviewer #2

(Remarks to the Author)

Dear Authors,
Dear Editor,
please finds my comments enclosed
Best regards

Version 1:

Reviewer comments:

Reviewer #1

(Remarks to the Author)

In this revision, the authors have appropriately addressed the comments and suggestions raised by the reviewers. I support the publication of this revision.

Reviewer #2

(Remarks to the Author)

Dear Authors,

Dear Editor,

please find attached my second round of (minor) comments

Best

Response to Reviewers

Reviewer #1

Reviewer's Comments:

In this study, the authors have applied metadynamics-biased *ab initio* molecular dynamics (AIMD) simulations with high-level quantum chemistry calculations and reaction density functional theory to investigate the reaction mechanisms of HMS formation at the interface. This study does provide a new perspective on the role of heterogeneous reactions on the HMS formation, advancing our understanding of atmospheric chemistry.

Response: We thank the reviewer for the positive evaluation of the advancement of understanding of atmospheric chemistry in the work.

Reviewer's Comments:

I did appreciate the authors have carried out very thorough and detailed model simulations. Overall, the presentation of the results are clear and the discussion concise and thoughtful. The proposed mechanisms are supported by the simulations and arguments are reasonable. In particular, the authors demonstrate the potential role of water molecules at the interface in the reactions and formation of HMS. This is a very nice chemistry and modeling paper. Indeed, I don't have specific comment on the data and discussion (the manuscript can be published nearly in current form).

Response: We thank the reviewer for the positive evaluation regarding the results of the simulations and the discussion of the work. We appreciate the reviewer for suggesting this paper for publication in *Nature Communications*.

Reviewer's Concern:

I did appreciate the authors have carried out very thorough and insightful model simulations. However, my main question (or concern) is that the authors have only considered relative simple systems. How their results and findings could be applied for more complex aerosol mixtures and cloud droplets. For instance, in addition to aerosol acidity or pH, how the presence of other species (e.g. inorganic ions and organic compounds (e.g. organic surfactants)) affect the reaction rates or mechanisms at the interface or bulk solution phase? How the composition and concentration of the species would alter the reaction rates and mechanisms? I thought this would be important for the authors to address or comment such that the findings of this work could have boarder and greater applications.

Response: We appreciate the reviewer for the very positive evaluation of our work and for the constructive comments to broaden the applicability of our manuscript. To evaluate the effect of inorganic ions on the reaction energy profile in the bulk solution, we have conducted new metadynamics-biased *ab initio* molecular dynamics (AIMD) simulations for the reaction between bisulfite (HOSO_2^-) and formaldehyde (HCHO) in the presence of sodium chloride (NaCl) as a typical inorganic ion. Compared to the pure bulk solution, the addition of salt ion leads to a slight decrease in the free-energy barrier but does not alter the reaction mechanism. We have further investigated the role of pH by exploring the contribution of sulfonate (HSO_3^-) in aerosol reactions under moderate acidic conditions (pH = 2–4). The simulation results identify the reaction between HSO_3^- and HCHO at the

air-water interface as a supplementary minor pathway, due to a much higher reaction energy barrier, compared to the reaction $\text{HOSO}_2^- + \text{HCHO}$. The schematic representation in **Fig. 7a** of the original manuscript has been revised accordingly to reflect this insight. Reaction mechanisms involving sulfonate as the predominant species under extreme acidic conditions have been included in the Supplementary Information (**Figs. S27 and S28**).

NB: Considering the presence of sulfonate as the isomer of bisulfite in atmospheric organic aerosols, particularly in acidic aerosols, we have revised the chemical representations of the relevant species to reflect accurate molecular configurations. Due to the hydrogen atom in the bisulfite bonded to the oxygen atom, we have modified the chemical formula of bisulfite to HOSO_2^- , whereas the isomer sulfonate (HSO_3^-) features the hydrogen atom bonded to the sulfur atom. These revised notations will be consistently applied throughout the revised manuscript.

1) Influence of the Inorganic Ions

The metadynamics-biased AIMD simulation were performed to investigate the reaction mechanism between bisulfite (HOSO_2^-) and HCHO in the presence of Na^+ and Cl^- ions (**Fig. R1**). The simulation was conducted using a cubic periodic boundary box ($1.42 \times 1.42 \times 1.42 \text{ nm}^3$), containing one Na^+ ion, one Cl^- ion, and 93 water molecules. The collective variable (CV) was defined as the distance between the sulfur atom of HOSO_2^- and the carbon atom of HCHO. The simulation system was carried out in the canonical (NVT) ensemble at the room temperature of 298.15 K, maintained via the velocity rescaling thermostat (CSVR) method. The time step for the simulation was set to 1 fs.

The representative snapshots from the AIMD trajectory (**Fig. R1a**) indicate that the inclusion of the inorganic ion does not alter the nucleophilic addition mechanism between HOSO_2^- and HCHO relative to the ion-free aqueous environment (see **Fig. 2d** in the original manuscript). The reaction still proceeds via a stepwise pathway involving the nucleophilic attack followed by the proton transfer, culminating in the formation of HMS. The CV variation (blue lines in **Fig. R1c**) delineates the progression through the reactant state (\mathbf{R}_{ion}), the transition state (\mathbf{TS}_{ion}), and the final state (\mathbf{P}_{ion}), corresponding to the structural evolution in **Fig. R1a**. Furthermore, the Na^+ ion (purple ball) remains solvated by six water molecules (licorice style), forming a well-defined first hydration shell. The distance (D_{ion}) between the Na^+ ion and the center-of-mass (COM) of reactants, illustrated by the purple line in **Fig. R1c**, fluctuates around 5.5–8 Å, with a spatial separation of approximately two hydration layers. These observations reveal that Na^+ ion acts as a non-interacting spectator, exerting only indirect solvation-mediated effects on the reaction free-energy profile.

Fig. R1. The reaction between HOSO_2^- and HCHO in the bulk salt solution. (a) Snapshot structures (reactant, R_{ion} , transition state, TS_{ion} and product, P_{ion}) obtained from the metadynamics-biased AIMD simulations. The purple ball indicates the Na^+ ion and the green ball represents the Cl^- ion. The water molecules coordinated to the Na^+ ion are marked as the licorice style for clarity. **(b)** Gibbs free-energy profile as a function of collective variable (CV). **(c)** (Left) Temporal evolution of the distance between the Na^+ ion and the COM (star) of reactants (D_{ion} , purple), along with the variation of the CV (blue). (Right) Schematic definition of D_{ion} and the CV.

From the view of chemical dynamics, Zhang *et al.* [*Atmos. Environ.* 294, 119474 (2023)] reported a quantitative relationship between the reaction rate constant of the HMS formation and the ionic strength in bulk solution. At low ionic concentrations, the relationship is expressed by equation (1)

$$\lg k = \frac{17.59I}{8.52 + I} + 0.57 \quad (1)$$

where I represents the ionic strength and k denotes the reaction rate constant. The ionic strength is calculated by the following formula:

$$I = \frac{1}{2} \sum c_i z_i^2 \quad (2)$$

in which c_i is the molar concentration (mol/L) of the ion i and z_i is its charge. For the reaction in pure water, the ionic strength (I_{bulk}) is effectively zero. In contrast, for the salt solution in our additional simulation system, the salt concentration is equal to 0.577 mol/L, yielding the ionic strength (I_{ion}) of 0.29 mol/L. Substituting into the equation (1), the ratio of the rate constant between the pure water and salt solution ($k_{\text{bulk}}/k_{\text{ion}}$) is calculated to be 0.264. Combining this value with the Eyring equation:

$$k = \frac{\kappa k_{\text{B}} T}{h} e^{-\frac{\Delta G^\ddagger}{RT}} \quad (3)$$

where the rate constant is determined by the activation energy ΔG^\ddagger , κ is the transmission coefficient, k_{B} is the Boltzmann constant, T is the temperature, R is the universal gas constant, and h is the Planck

constant. Based on this formalism, the difference in the Gibbs free-energy barrier ($\Delta G_{\text{bulk}}^{\ddagger} - \Delta G_{\text{ion}}^{\ddagger}$) can be calculated to be approximately 0.8 kcal/mol. As shown in **Fig. R1b**, the $\Delta G_{\text{ion}}^{\ddagger}$ of the salt solution system is determined to be 10.2 kcal/mol, while the $\Delta G_{\text{bulk}}^{\ddagger}$ of the pure water system is ~ 11.4 kcal/mol, yielding an energy barrier difference of 1.2 kcal/mol. The close agreement between theoretical predictions and experimental estimations substantiates the proposed reaction mechanism and underscores the role of ionic strength in modulating reaction kinetics.

2) Influence of Aerosol Acidity or pH

Numerous experiments based on field observations [*Atmos. Chem. Phys.* 23, 10795-10807 (2023), *NPJ Clim. Atmos. Sci.* 6, 173 (2023), *Sci. Adv.* 10, eado4373 (2024) and *Nat. Commun.* 15, 8987 (2024)] have highlighted the role of aerosol acidity (pH) in modulating the kinetics of HMS formation. For example, Buttersack *et al.* [*Nat. Commun.* 15, 8987 (2024)] employed X-ray photoelectron spectroscopy (XPS) and Raman spectroscopy to identify the isomer conversion between the sulfonate (HSO_3^-) and bisulfite (HOSO_2^-) at the acidic air-water interface. Notably, the sulfonate form becomes to be predominant species under extreme acidic conditions (pH = 0.8).

Hong *et al.* [*Atmos. Chem. Phys.* 23, 10795-10807 (2023)] and Huang *et al.* [*NPJ Clim. Atmos. Sci.* 6, 173 (2023)] have investigated the HMS formation at moderate acidic organic aerosol (pH = 2–4). The statistical analysis by Hong *et al.* [*Atmos. Chem. Phys.* 23, 10795-10807 (2023)] revealed that the HMS formation is enhanced under moderate acidic conditions (pH = 3–4). Additionally, the XPS measurement performed by Buttersack *et al.* [*Nat. Commun.* 15, 8987 (2024)] confirmed the coexistence of the HSO_3^- and HOSO_2^- at air-water interface, where HOSO_2^- can react with HCHO to form HMS. Furthermore, Campbell *et al.* [*Sci. Adv.* 10, eado4373 (2024)] emphasized the contribution of sulfite (SO_3^{2-}) at weak acidic conditions (pH > 4), facilitating the HMS production.

Consequently, in order to explore the underlying reaction pathway at extreme acidic conditions (pH = 0.8–1.8), we have employed AIMD simulations and quantum chemical calculations for the reaction $\text{HSO}_3^-(aq.) + \text{HCHO}(aq.) \rightarrow \text{HOCH}_2\text{OSO}_2^-$, forming the HMS isomer (HMSi). Building on the AIMD calculations of the $\text{HOSO}_2^-(aq.) + \text{HCHO}(aq.) \rightarrow \text{HOCH}_2\text{SO}_3^-$ and $\text{SO}_3^{2-}(aq.) + \text{HCHO}(aq.) \rightarrow \text{O}^-\text{CH}_2\text{SO}_3^-$ reactions, the results provide mechanistic insights into the interface and bulk solution phases across varying pH conditions, as illustrated in **Figs. R2 to R5**.

a) Reaction mechanism at extreme acidic conditions (pH = 0.8–1.8)

At extreme acidic conditions, sulfonate (HSO_3^-) becomes the predominant species at air-water interface [*Nat. Commun.* 15, 8987 (2024)], and its reactivity of HSO_3^- with HCHO is presented in **Fig. R2**. The electrostatic potential map of HSO_3^- , as shown in **Fig. R2a**, reveals a pronounced negative potential (blue region) around the oxygen of HSO_3^- , signifying that the oxygen of HSO_3^- serves as the primary nucleophilic site. In contrast, the hydrogen atom bonded to the sulfur atom

carries a partial positive charge, indicating a propensity for the proton transfer from the sulfur-bound H to electrophilic centers such as the carbonyl oxygen of HCHO. Frontier molecular orbital analysis reveals that the highest occupied molecular orbital (HOMO) of HSO_3^- is predominantly localized on the oxygen atom, reinforcing the nucleophilic character of the lone pair p electron on the oxygen atom.

Fig. R2. Reactivity of the S(IV) species at extreme acidic conditions (pH = 0.8–1.8). (a) Electrostatic potential surface of HSO_3^- , with red denoting regions of positive electrostatic potential and blue regions indicating negative potential. (b) Isovalue surfaces of HOMO orbital (isovalue = ± 0.005) of the HSO_3^- . (c) A representative mechanism for $\text{HCHO} + \text{HSO}_3^-$ reaction, where the HSO_3^- functions as the nucleophile to attack the carbonyl group. (d) Gibbs free-energy profiles for $\text{HCHO} + \text{HSO}_3^-$ reaction in gas phase and their corresponding structures of the stationary points (**R**, **INT**, **TS** and **P**) optimized at M06-2x/6-311++G(d,p) level of theory. (e) Gibbs free-energy profile of the $\text{HCHO} + \text{HSO}_3^-$ reaction at air-water interface obtained from TI-AIMD simulation with the average values of each window (blue dots) for three independent simulations, the error bar (colored orange) and the interpolation line (red line) of PCHIP method. (f) (Top) Definition of the CV for the TI-AIMD simulation. (Bottom) Snapshot structures (reactant, **R_{sul}**, transition state, **TS_{sul}** and product, **P_{sul}**) obtained from the TI-AIMD simulations.

In conjunction with the electronic structure analysis of HCHO in **Figs. 4b** and **4c** of the original manuscript, the nucleophilic addition mechanism of between the HSO_3^- and the HCHO is illustrated in **Fig. R2c**. The reaction proceeds via a two-step mechanism: the nucleophilic addition from the oxygen in HSO_3^- to the carbonyl in HCHO, followed by the intramolecular proton transfer. The resulting product is an isomer of HMS (HMSi). **Fig. R2d** depicts the gaseous reaction of $\text{HSO}_3^- + \text{HCHO}$, where the reactant needs to overcome a free-energy barrier of $\Delta G^\ddagger = 11.3$ kcal/mol to

form an intermediate (INT). For transition state (TS), HSO_3^- and HCHO forms a five-membered ring stabilized by both the hydrogen bonding $[\text{H}(\text{HSO}_3^-) \cdots \text{O}(\text{HCHO})]$ and the nucleophilic interaction $[\text{S}(\text{HSO}_3^-) \cdots \text{C}(\text{HCHO})]$. This arrangement facilitates the intramolecular proton transfer from the sulfur-bound hydrogen to the carbonyl oxygen of HCHO. However, the significant ring strain associated with this geometry of TS results in a relatively high free-energy barrier of $\Delta G^\ddagger = 53.6$ kcal/mol.

To examine the interfacial reactivity, the free-energy profile of HCHO + HSO_3^- reaction at the air-water interface was obtained using the thermal integration (TI)-AIMD method. The simulation was employed by a water slab comprising 96 water molecules in a periodic box of dimensions $1.5 \times 1.5 \times 3.0$ nm³. The CV was defined as the distance between the oxygen atom in HSO_3^- and the carbon atom in HCHO (Fig. R2e). A total of 11 sampling windows were implemented to guarantee the smoothness of the calculated free-energy profile. Each window was equilibrated for ~15 ps, followed by a production run of 5 ps for the free-energy sampling. All simulations were performed in the canonical (NVT) ensemble at 298.15 K with temperature control maintained using the velocity rescaling thermostat (CSVR) method. The time step for the simulation was set to 1 fs.

As shown in Fig. R2e, the blue dots represent the average value obtained from three independent simulations, while the free-energy profile (red line) is generated by Piecewise Cubic Hermite Interpolating Polynomial (PCHIP) interpolation method. The error bars for each average value are colored orange. The results show that the free-energy barrier ($\Delta G_{\text{sul}}^\ddagger$) of the HCHO + HSO_3^- reaction at the air-water interface is 18.1 ± 1.0 kcal/mol. The snapshot structures of the interfacial HCHO + HSO_3^- reaction are provided in Fig. R2f.

Overall, as schematically illustrated in Fig. R3, these results show that at extreme acidic condition (pH = 0.8–1.8) where the sulfonate (HSO_3^-) is the dominant aqueous S(IV) species, the surface-accumulated HCHO can undergo nucleophilic addition with this species to form HMSi.

Fig. R3. The role of sulfonate (HSO_3^-) under extreme acidic conditions (pH = 0.8–1.8). The abundant presence of HSO_3^- at the air-water interface enables its reaction with surface-accumulated HCHO to form HMSi.

b) Reaction mechanism at moderate acidic conditions (pH = 2–4)

Fig. R4 summarizes the reaction pathways involving various S(IV) species under moderate acidic organic aerosols (pH = 2–4). At such condition, the HSO_3^- can undergo tautomerization to form HOSO_2^- with an equilibrium constant $K_T = 3.2$ at pH = 4 [Nat. Commun. **15**, 8987 (2024)]. Although the HSO_3^- remains the dominant S(IV) species at the acidic interface, the metadynamics-biased AIMD simulation results reveal that the nucleophilic addition between HOSO_2^- and HCHO proceeds with a significantly lower energy barrier ($\Delta G^\ddagger = 7.6$ kcal/mol) compared to the $\text{HCHO} + \text{HSO}_3^-$ reaction ($\Delta G_{\text{sul}}^\ddagger = \sim 18.1$ kcal/mol). This indicates that, despite the relative lower concentration compared to HSO_3^- , HOSO_2^- exhibits substantially higher reactivity toward the surface-accumulated HCHO , thereby accelerating the formation of HMS. Moreover, the tautomeric equilibrium between HSO_3^- and HOSO_2^- enables a continuous supply of the more reactive HOSO_2^- species, sustaining the overall conversion process. This dynamic equilibrium underpins a cascade reaction mechanism, ensuring efficient HMS production under moderate acidic environment.

Fig. R4. Reaction pathways for S(IV) species at acidic organic aerosols (pH = 2–4). Despite sulfonate (HSO_3^-) predominates in the equilibrium with bisulfite (HOSO_2^-) under moderate acidic conditions, the reaction between HOSO_2^- and surface-accumulated HCHO at the air-water interface proceeds with a relatively low energy barrier ($\Delta G^\ddagger = 7.6$ kcal/mol) compared to the HMSi formation ($\Delta G_{\text{sul}}^\ddagger = 18.1$ kcal/mol). This enables the cascade formation of HMS, depleting the concentration of HOSO_2^- , and driving the ongoing conversion of HSO_3^- to HOSO_2^- . As a result, HOSO_2^- continuously reacts with HCHO to generate HMS, sustaining the HMS formation.

c) Reaction mechanism from weak acidic to neutral conditions (pH > 4)

Fig. R5 outlines the reaction pathways concerning S(IV) species from weak acidic to neutral organic aerosols (pH > 4). As reported by Buttersack *et al.* [Nat. Commun. **15**, 8987 (2024)], the concentration of SO_3^{2-} increases significantly at pH > 4, primarily due to the deprotonation of HSO_3^- ($pK_{\text{SH}} = 6.3$) and HOSO_2^- ($pK_{\text{OH}} = 6.9$) [Nat. Commun. **15**, 8987 (2024)]. In such condition, the formation of HMS is energetically favorable, with low energy barrier of $\text{SO}_3^{2-} + \text{HCHO}$ reaction both at the air-water interface ($\Delta G^\ddagger = \sim 2.7$ kcal/mol) and in the aqueous solution ($\Delta G^\ddagger = \sim 5.8$

kcal/mol).

Fig. R5. Reaction pathways concerning S(IV) species from weak acidic to neutral organic aerosols (pH > 4). The concentration of sulfite (SO_3^{2-}) augments due to the deprotonation of the sulfonate (HSO_3^-) and the bisulfite (HOSO_2^-). The SO_3^{2-} reacts with HCHO at the air-water interface or in the bulk solution to form the HMS.

In conclusion, the aforementioned study provides a comprehensive mechanistic understanding of how the inorganic ions and aerosol acidity modulate the HMS formation under atmospheric conditions. These insights have been integrated into the revised manuscript under the section entitled ‘**Quantitative Mechanism of the Heterogeneous HMS Formation**’ within the manuscript. Finally, we would like to extend our sincere appreciation to the reviewer for the constructive comments, which has significantly enhanced the clarity and depth of our work.

Revision:

We have added two discussions in the revised manuscript:

*In moderate acidic aerosol (pH = 2–4), the HMS is formed by the nucleophilic addition by HCHO and HOSO_2^- . Although the tautomeric equilibrium of $\text{HOSO}_2^- \rightleftharpoons \text{HSO}_3^-$ ($K_T = 3.2$ at pH = 4) leads to a relative low concentration of HOSO_2^- at the air-water interface. According the reactivity of HSO_3^- (**Supplementary Note 13** and **Fig. S27**), the free-energy barrier ($\Delta G_{\text{su}}^\ddagger$) of the $\text{HCHO} + \text{HSO}_3^-$ reaction at the air-water interface is 18.1 ± 1.0 kcal/mol, which is significantly higher than that of $\text{HOSO}_2^- + \text{HCHO}$ reaction ($\Delta G^\ddagger = \sim 7.6$ kcal/mol). As a result, compared to HSO_3^- , HOSO_2^- exhibits substantially higher reactivity toward the surface-accumulated HCHO, thereby accelerating the formation of HMS. Moreover, the tautomeric equilibrium between HSO_3^- and HOSO_2^- enables a continuous supply of the more reactive HOSO_2^- species, sustaining the overall conversion process. This dynamic equilibrium underpins a cascade reaction mechanism, ensuring efficient HMS production under moderate acidic environment.’ in the revised manuscript [p. 17, para. 2, lines 18–28].*

*‘Additionally, at extreme acidic condition (pH = 0.8–1.8, **Supplementary Note 14** and **Fig. S28**),*

Buttersack et al. reported that HSO_3^- becomes the predominant species at the air-water interface. The surface-accumulated HCHO can undergo nucleophilic addition with this species to form HMS isomer. As the pH increases to weak acidic or neutral conditions ($\text{pH} > 4$), the concentration of SO_3^{2-} rises significantly. Under such condition, the formation of HMS is energetically favorable, due to a low energy barrier of $\text{SO}_3^{2-} + \text{HCHO}$ reaction both at the air-water interface and in the aqueous solution (Supplementary Note 14 and Fig. S29). Furthermore, the effect of inorganic ions on the reaction energy profile in the bulk solution is investigated. The free-energy barrier for the $\text{HOSO}_2^- + \text{HCHO}$ reaction in the salt solution ($\Delta G_{\text{ion}}^\ddagger$) is determined to be 10.2 kcal/mol (Supplementary Note 12 and Fig. S26). Compared to the pure bulk solution ($\Delta G_{\text{bulk}}^\ddagger = 11.4$ kcal/mol), the presence of salt ion slightly lowers the free-energy barrier but does not alter the underlying reaction mechanism.’ in the revised manuscript [p. 19, para. 2, lines 13–23].

More discussion has been incorporated into the revised Supplementary Information [p. 45–52, Note 12–14, Figs. S26–S29].

Reviewer #2

Reviewer’s Concern:

The authors report a computational study on the formation of hydroxymethanesulfonate (HMS) from bisulfite, HSO_3^- (or sulfite, SO_3^{2-}) and formaldehyde, HCHO, at the aqueous-gas interface under acidic (or basic) conditions, respectively. As stated in the introduction, HMS is a crucial component in haze formation in polluted atmospheres. Compared to the gas-phase reaction or the reaction in homogeneous liquid water, this study reports a 563-fold enhancement of the reaction rate at the interface in acid conditions.

The authors have conducted an impressive amount of work. However, several technical and fundamental aspects are lacking or questionable, which undermines the study. Key concerns (more details are reported below):

1. The metadynamics (MTD) simulations do not appear to have reached convergence. Since reaction rates depend exponentially on reaction barriers, the conclusions regarding accelerated rates are difficult to be supported.
2. Sulfonate, rather than bisulfite, is typically present at the surface of acid liquid droplets, limiting the implications of the reported findings.
3. Some parts of the text require better clarity.

Unfortunately, based on these issues, I am sorry but I cannot support the publication of this work.

Response: We thank the reviewers for the positive evaluation, particularly for pointing out ‘*the impressive amount of the work*’. We sincerely appreciate the time and effort the reviewer dedicated to providing constructive feedback, which has significantly strengthened the quality of our manuscript. Below, we have carefully provided a point-by-point response addressing each concern in detail:

1. We totally agree with the reviewer's concern regarding the need to validate the convergency of the metadynamics-biased AIMD simulation. To address this, we have employed additional simulation using the thermal integration (TI)-AIMD simulation. We find that the free-energy barriers obtained from metadynamics and TI-AIMD simulations are consistent within the margin of error, thereby reinforcing the reliability of the metadynamics simulations and confirming the reaction acceleration effect at the air-water interface.
2. In response to the reviewer's inquiry regarding the isomer form of bisulfite, we have expanded our discussion to include the reaction between sulfonate (HSO_3^-) and HCHO at air-water interface. The additional simulation results show that the free-energy barrier for the $\text{HSO}_3^- + \text{HCHO}$ reaction at air-water interface is relative higher than that for the $\text{HOSO}_2^- + \text{HCHO}$ reaction at air-water interface. Consequently, the $\text{HSO}_3^- + \text{HCHO}$ reaction may contribute to HMS isomer formation under moderate acidic condition, but its role is likely secondary compared to the HOSO_2^- pathway.
3. For the reviewer's comments concerning the clarity of some parts of the text in the original manuscript, we have carefully revised the manuscript throughout to enhance readability and ensure that the scientific narrative is communicated more effectively.

We hope that these revisions and additional analyses adequately address the reviewer's concerns and further substantiate the robustness and significance of our findings.

Reviewer's Concern:

Clarity:

For future submissions, please include page and line numbers in your draft. This will help reviewers to easily reference specific parts of the manuscript.

Response: We sincerely appreciate the referee for the comment, and we fully understand the importance of this feature in facilitating a smooth and efficient review process. Moving forward, we will ensure that all future submissions include page and line numbers to make it easier for reviewers to reference specific sections of the manuscript. We appreciate the reviewer's suggestion and will implement this change immediately.

Reviewer's Concern:

The Abstract

The abstract is somewhat difficult to follow:

Response: We sincerely appreciate the referee for the comment. In response, we have carefully revised the abstract to enhance its clarity.

Reviewer's Concern:

a) "...compared to gaseous HMS formation, the solvated reactants ..." Does "solvated reactants" refer to those in the homogeneous bulk solution or at the air-water interface? Please clarify.

Response: We apologize for the unclarity in the original manuscript and sincerely appreciate the referee for the comment. In this context, the ‘solvated reactants’ refers to the reactants in the homogeneous bulk solution. Based on the reviewer’s concern, we have modified the phrase ‘solvated reactants’ to ‘*reactants in the homogeneous bulk solution*’.

Reviewer’s Concern:

Additionally, before discussing the individual contributions (i.e., polarization effects, water reorganization, etc.) to the reaction barriers, clearly state the reaction barrier values at the interface, in homogeneous bulk and gas conditions. This information is crucial to immediately grasp the catalytic effect of the heterogeneous environment.

Response: We sincerely appreciate the referee for the comment. We have emphasized this aspect in the abstract and conclusion to highlight the effect of heterogeneous catalysis in the revised manuscript.

In the abstract, we have revised the original statement as follows:

‘RxDFT calculations indicate that, compared to gaseous HMS formation, the reactants in the *homogeneous bulk solution* should overcome a water reorganization barrier of ~5.0 kcal/mol while partly compensated by water polarization effect (~2.1 kcal/mol). Interestingly, according to the AIMD simulation, the cooperative hydrogen bonding networks of water molecules bridge the loop of proton transfer channel through Grotthuss mechanism to reduce the activation entropy by ~5.5 kcal/mol, thereby lowering *the free-energy barrier in the bulk solution* to ~11.4 kcal/mol, in close agreement with the experimental result.’

Similarly, in the conclusion, we have modified the original statement as follows:

‘*In the bulk solution*, according to the RxDFT results, compared to the gas-phase reaction, solvated reactants must overcome a water reorganization barrier of approximately 5.0 kcal/mol, partially compensated by a polarization effect of ~2.1 kcal/mol. Notably, the cooperative hydrogen-bonding networks of water molecules facilitate proton transfer via the Grotthuss mechanism, reducing activation entropy by approximately 5.5 kcal/mol. This lowers the free-energy barrier *in the bulk solution* to ~11.4 kcal/mol, closely agreeing with experimental data. At the air-water interface, the partial solvation of HCHO further decreases the activation enthalpy by ~1.0 kcal/mol. Additionally, the interfacial stabilization arising from restricted translation and rotation of HCHO reduces the configurational entropy barrier by ~0.9 kcal/mol. This synergistic regulation of entropy and enthalpy effect is comparable to interfacial electric field effect with a reduction of 1.9 kcal/mol, leading to a remarkable two orders of magnitude enhancement at the air-water interface rate compared to the aqueous reaction.’

This revision highlights the contributions of various effects to heterogeneous catalysis, aligning with the data presented in **Fig. 7** of the revised manuscript.

Reviewer’s Concern:

b) The abstract is somewhat misleading because it implies that ab-initio molecular dynamics (MD) and free energy profiles were obtained using RxDFT, while they were actually computed using BLYP-D3 DFT (as stated in the Methods section).

Response: We sincerely appreciate the referee for the comment, we have explicitly distinguished this aspect in the revised abstract as follows:

'RxDFT calculations indicate that, compared to gaseous HMS formation, the reactants in the homogeneous bulk solution should overcome a water reorganization barrier of ~ 5.0 kcal/mol while partly compensated by water polarization effect (~ 2.1 kcal/mol). Interestingly, *AIMD simulation shows that* the cooperative hydrogen bonding networks of water molecules bridge the loop of proton transfer channel through Grotthuss mechanism to reduce the activation entropy by ~ 5.5 kcal/mol, thereby lowering the free-energy barrier in the bulk solution to ~ 11.4 kcal/mol, in close agreement with the experimental result.'

Reviewer's Concern:

The Introduction

"In this atmospheric acidic condition, the dissolved S(IV) predominately exists as HSO_3^- . Therefore, this pose a contradiction that arises from the existing high concentration of HMS in the acidic aqueous environment with the abundant HSO_3^- , while the reaction rate of HCHO with HSO_3^- in solution phase is five orders of magnitude lower than that with SO_2^- . "

This sentence is a bit unclear. I think "while" should be replace by "because"

Response: We sincerely appreciate the referee for the suggestion. The 'while' has been replaced by '*because*' in the revised manuscript.

*'Therefore, this pose a contradiction that arises from the existing high concentration of HMS in the acidic aqueous environment with the abundant HSO_3^- , **because** the reaction rate of HCHO with HSO_3^- in solution phase is five orders of magnitude lower than that with SO_3^{2-} . '*

Reviewer's Concern:

Bisulfite at acid aqueous environment.

The authors investigate the reactions between

- a) bisulfite (HSO_3^-) and formaldehyde (HCHO) \rightarrow HMS (acid conditions)
- b) sulfite (SO_3^{2-}) and formaldehyde (HCHO) \rightarrow HMS. (basic conditions)

in the gas phase, in bulk aqueous solution, and at the air-water interface. Reaction b) is relevant at basic conditions, Reaction a) at low pH.

Regarding Reaction a), the authors state in the abstract: “Collectively, these factors lead to a remarkable 563-fold enhancement in the reaction rate at the air-water interface compared to the aqueous reaction.” This is presented as one of the key findings of the paper, as seen in the abstract.

However, a recent Nature Communications study (Ref. 77 in the manuscript) combined XPS, Raman spectroscopy, and well-converged ab initio metadynamics (MTD) simulations to demonstrate that sulfonate, rather than bisulfite, is the most abundant and stable S(IV) species at the surface of an acidic air-water interface. It is surprising that, despite citing Ref. 77, the authors did not account (or at least discussed) for this finding, and focused their study on bisulfite. Indeed, in their conclusion, they state: “Our findings reveal that, although HSO_3^- is the dominant sulfur species in acidic environments...” However, given the evidence from Ref. 77, the conclusions about the interfacial reaction (at least for the acid conditions) starting from bisulfite and HCHO have questionable relevance and impact.

Response: We sincerely appreciate the referee for your insightful comment, which have significantly enhanced the rigor and comprehensiveness of our discussion. In our original manuscript, we inadvertently ignored the presence of sulfonate at air-water interface. As a result, we have performed additional quantum chemistry calculation and AIMD simulation to assess the role of sulfonate in the aerosol reactions under moderate acidic conditions (pH = 2–4), where it may sever as a supplementary, albeit minor, pathway for HMS isomer formation. Based on this new result, we have revised the schematic representation in **Fig 7a** of the original manuscript to reflect this additional reaction channel. Furthermore, the reaction mechanisms involving sulfonate at extreme acidic conditions have been included in the Supplementary Information (**Figs. S27 and S28**).

NB: In light of the coexistence of bisulfite and its isomer sulfonate in atmospheric organic aerosols, particularly under acidic aerosols, we have revised the chemical representations of the relevant species to reflect their distinct molecular structures. Due to the hydrogen atom in the bisulfite bonded to the oxygen atom, we have modified the chemical formula of bisulfite to HOSO_2^- , whereas the isomer sulfonate (HSO_3^-) features the hydrogen atom bonded to the sulfur atom. These updated representations have been applied consistently throughout the revised manuscript and Supplementary Information to enhance chemical clarity and precision.

1) Investigation of reactivity of HSO_3^-

The reactivity of HSO_3^- with HCHO is presented in **Fig. R6**. Firstly, the electronic structure of the HSO_3^- is analyzed in **Figs. R6a and R6b**. As shown in **Fig. R6a**, the blue region around the oxygen atom of HSO_3^- indicates a negative electrostatic potential, signifying that the oxygen atom of HSO_3^- serves as the nucleophilic site. Moreover, the hydrogen atom bonded to the sulfur atom carries a positive charge, which suggests the potential pathway for the proton transfer from the sulfur-bound H to the carbonyl atom. As shown in **Fig. R6b**, the electron distribution in highest occupied molecular orbital (HOMO) of HSO_3^- is mainly localized around the oxygen atom, confirming the nucleophilicity character of the lone pair electrons on the oxygen during the reaction with HCHO.

Fig. R6. Accelerated reactivity of the sulfonate (HSO₃⁻) to form the HMSi formation at the air-water interface. (a) Electrostatic potential surface of HSO₃⁻. Red regions denote positive electrostatic potential and blue regions of negative potential. (b) Isovalue surfaces of the highest occupied molecular orbital (HOMO) of the HSO₃⁻ (isovalue = ±0.005). (c) Representative reaction mechanism for HCHO + HSO₃⁻ reaction, where the HSO₃⁻ functions as the nucleophile to attack the carbonyl group. (d) Gibbs free-energy profiles for HCHO + HSO₃⁻ reaction in gas phase with their corresponding optimized structures of the stationary points (R, INT, TS and P) at the M06-2x/6-311++G(d,p) level of theory. (e) Gibbs free-energy profile of the HCHO + HSO₃⁻ reaction at the air-water interface obtained from TI-AIMD simulation with the average values of each window (blue dots) for three independent simulations, the error bar (colored orange) and the interpolation line (red line) of PCHIP method. (f) (Top) Definition of the CV for the TI-AIMD simulation. (Bottom) Snapshot structures (reactant, R_{sul}, transition state, TS_{sul} and product, P_{sul}) obtained from the TI-AIMD simulations.

Combined with the electronic structure analysis of the HCHO in Figs. 4b and 4c of the original manuscript, the mechanism of the nucleophilic addition between the HSO₃⁻ and the HCHO is illustrated in Fig. R6c. The reaction proceeds via a two-step mechanism: the nucleophilic addition from the oxygen in HSO₃⁻ to the carbonyl in HCHO, followed by the proton transfer. The product of this reaction is an isomer of HMS (HMSi). As shown in Fig. R6d, the gaseous reaction of HSO₃⁻ + HCHO begins with the formation of an intermediate (INT), requiring an initial free-energy barrier of ΔG[‡] = 11.3 kcal/mol. For transition state (TS), HSO₃⁻ and HCHO forms a five-membered ring through the hydrogen bonding [H(HSO₃⁻)...O(HCHO)] and nucleophilic interaction [S(HSO₃⁻)...C(HCHO)], initiating the intramolecular proton transfer from the sulfur-bound hydrogen to the carbonyl oxygen of HCHO. However, due to the considerable ring strain within the TS, the free-energy barrier is relatively high (ΔG[‡] = 53.6 kcal/mol).

To obtain the free-energy profile for $\text{HCHO} + \text{HSO}_3^-$ reaction at air-water interface, we employed the thermal integration (TI)-AIMD method. The simulations were performed on a water slab with two air-water interfaces, comprising 96 water molecules in a simulation box of dimensions $1.5 \times 1.5 \times 3.0 \text{ nm}^3$. The CV was defined as the distance between an oxygen atom in HSO_3^- and the carbon atom in HCHO (**Fig. R6f**). A total of 11 sampling windows were implemented to guarantee the smoothness of the calculated free-energy profile. Each window was subjected to ~ 15 ps of pre-equilibration for structural relaxation, followed by a 5 ps simulation run for the free-energy computation. All simulations were performed in the canonical (NVT) ensemble at 298.15 K using the velocity rescaling thermostat (CSVR) method. The time step for the simulation was set to 1 fs.

As shown in **Fig. R6e**, the blue dots represent the average value of each three simulations, and the free-energy surface (red line) is generated by Piecewise Cubic Hermite Interpolating Polynomial (PCHIP) interpolation method. Orange error bars indicate the standard deviation of each averaged data point. The calculated free-energy barrier ($\Delta G_{\text{sul}}^\ddagger$) of the $\text{HCHO} + \text{HSO}_3^-$ reaction at air-water interface is $18.1 \pm 1.0 \text{ kcal/mol}$ much higher than that of $\text{HCHO} + \text{HOSO}_2^-$ reaction under similar conditions. The snapshot structure of the $\text{HCHO} + \text{HSO}_3^-$ reaction is displayed in **Fig. R6f**.

2) Role of HSO_3^- in organic aerosol under acidic conditions

According to the experimental result reported by Buttersack *et al.* [*Nat. Commun.* **15**, 8987 (2024)], the HSO_3^- is the dominant species at the air-water interface under acidic conditions. However, the ratio of species is highly pH-dependent. For example, HOSO_2^- is nearly absent under extreme acidic condition ($\text{pH} = 0.8\text{--}1.8$), whereas at moderately acidic conditions ($\text{pH} = 2\text{--}4$), a chemical equilibrium between HSO_3^- and HOSO_2^- is established at the air-water interface. Therefore, a differentiated mechanistic analysis is warranted for these two distinct scenarios.

First, the interfacial reaction mechanism at extreme acidic condition ($\text{pH} = 0.8\text{--}1.8$) is depicted in **Fig. R7**. Combined with the results in **Fig. R6**, the predominance of sulfonate (HSO_3^-) enables its reaction with surface-accumulated HCHO to form HMSi , with a calculated free-energy barrier of $\Delta G_{\text{sul}}^\ddagger = \sim 18.1 \text{ kcal/mol}$.

Fig. R7. The role of sulfonate (HSO_3^-) under extreme acidic conditions ($\text{pH} = 0.8\text{--}1.8$). The abundant

presence of HSO_3^- at the air-water interface enables its reaction with surface-accumulated HCHO to form HMSi.

Furthermore, **Fig. R8** illustrates the interfacial reaction mechanism at moderate acidic condition ($\text{pH} = 2-4$). At such condition, HSO_3^- can undergo tautomerization to form HOSO_2^- [*Nat. Commun.* **15**, 8987 (2024)], with an equilibrium constant $K_T = 3.2$ at $\text{pH} = 4$. Although the HSO_3^- remains the dominant S(IV) species under this acidic condition, the free-energy barrier ($\Delta G^\ddagger = \sim 7.6$ kcal/mol) of $\text{HCHO} + \text{HOSO}_2^-$ reaction at the air-water interface is markedly lower than that of the $\text{HCHO} + \text{HSO}_3^-$ reaction ($\Delta G_{\text{sul}}^\ddagger = \sim 18.1$ kcal/mol). This indicates that, compared to HSO_3^- , HOSO_2^- exhibits much higher reactivity toward surface-accumulated HCHO, leading to the more efficient formation of HMS. As a result of the tautomerization equilibrium, HSO_3^- serves as a dynamic reservoir that can be continuously converted into HOSO_2^- . The HOSO_2^- produced in this way subsequently reacts with HCHO, sustaining the formation of HMS under moderate acidic condition.

Fig. R8. The role of the sulfonate (HSO_3^-) under moderate acidic aerosols ($\text{pH} = 2-4$). Although HSO_3^- predominates in the equilibrium with bisulfite (HOSO_2^-) under acidic conditions, the reaction between HOSO_2^- and surface-accumulated HCHO at the gas-liquid interface proceeds with a relatively low energy barrier ($\Delta G^\ddagger = \sim 7.6$ kcal/mol) compared to the HMSi formation ($\Delta G_{\text{sul}}^\ddagger = \sim 18.1$ kcal/mol), leading to the cascade formation of HMS. This process depletes the concentration of HOSO_2^- , driving the ongoing conversion of HSO_3^- to HOSO_2^- . As a result, HOSO_2^- continuously reacts with HCHO to generate HMS, sustaining the HMS formation.

Revision:

We have revised the phrases in the revised manuscript:

'In this atmospheric acidic condition, the dissolved S(IV) exists as the tautomeric equilibrium between bisulfite (HOSO_2^-) and sulfonate (HSO_3^-).' in the revised manuscript [p. 2, para. 2, lines 35–36].

'Recent atmospheric field measurements have reported elevated HMS concentration in fog and cloud water under acidic conditions, where the ratio of S(IV) species is dependent on the tautomeric equilibrium between HOSO_2^- and HSO_3^- ' in the revised manuscript [p. 5, para. 1, lines 1–3].

We have added two discussions in the revised manuscript:

'In moderate acidic aerosol (pH = 2–4), the HMS is formed by the nucleophilic addition by HCHO and HOSO₂⁻. Although the tautomeric equilibrium of HOSO₂⁻ ⇌ HSO₃⁻ (K_T = 3.2 at pH = 4) leads to a relative low concentration of HOSO₂⁻ at the air-water interface. According the reactivity of HSO₃⁻ (Supplementary Note 13 and Fig. S27), the free-energy barrier ($\Delta G_{\text{su}}^\ddagger$) of the HCHO + HSO₃⁻ reaction at the air-water interface is 18.1 ± 1.0 kcal/mol, which is significantly higher than that of HOSO₂⁻ + HCHO reaction ($\Delta G^\ddagger = \sim 7.6$ kcal/mol). As a result, compared to HSO₃⁻, HOSO₂⁻ exhibits substantially higher reactivity toward the surface-accumulated HCHO, thereby accelerating the formation of HMS. Moreover, the tautomeric equilibrium between HSO₃⁻ and HOSO₂⁻ enables a continuous supply of the more reactive HOSO₂⁻ species, sustaining the overall conversion process. This dynamic equilibrium underpins a cascade reaction mechanism, ensuring efficient HMS production under moderate acidic environment.' in the revised manuscript [p. 17, para. 2, lines 18–28].

'Additionally, at extreme acidic condition (pH = 0.8–1.8, Supplementary Note 14 and Fig. S28), Buttersack et al. reported that HSO₃⁻ becomes the predominant species at the air-water interface. The surface-accumulated HCHO can undergo nucleophilic addition with this species to form HMS isomer.' in the revised manuscript [p. 19, para. 2, lines 13–15].

More discussion has been incorporated into the revised Supplementary Information [p. 48–51, Note 13–14, Figs. S27–S28].

Reviewer's Concern:

The Metadynamics

Figure 2f and Figure S3 clearly indicate that the metadynamics (MTD) simulations have not reached convergence. As shown in these figures, the collective variable (CV) exhibits only a single transition from reactants to products and never reaches a free-diffusive regime. There are no multiple recrossings, which is a necessary condition for well-converged MTD. The issue of MTD convergence has been widely discussed in the literature and across various scientific mailing lists, forum and tutorials (e.g., <https://www.plumed.org/doc-v2.9/user-doc/html/belfast-6.html>).

Response: We sincerely appreciate the referee for the comment. We fully acknowledge the importance of ensuring the convergence of metadynamics simulations by observing the multiple recrossings in the collective variable (CV) profile.

In our metadynamics simulations, a constant Gaussian bias potential was applied, with a relatively small amplitude of the Gaussian potential (1×10^{-4} Hartree, equivalent to 0.06 kcal/mol) compared to the reaction energy barrier (~ 10 kcal/mol). Additionally, the CV sampling space was relatively large (~ 3 Å), allowing for adequate exploring of the reaction pathway while balancing computational accuracy and efficiency. Consequently, each metadynamics run proceeded until the CV evolved from the reactant state, traversed the transition state, and reached the product state.

To further address the potential limitations of the metadynamics simulations, we have performed additional simulations using thermodynamic integration (TI)-AIMD simulation method for the reaction between SO_3^{2-} and HCHO in the bulk solution, as shown in **Fig. R9**. The simulation results show that our metadynamics results ($\Delta G_{\text{bulk}}^\ddagger = 5.8 \pm 0.1$ kcal/mol) is consistent with the free-energy barrier ($\Delta G_{\text{TI}}^\ddagger$) of 6.4 ± 0.7 kcal/mol by using (TI)-AIMD method, falling within the margin of error. This agreement confirms the robustness of the metadynamics approach and validates the reliability of reaction energetics derived from it.

Reviewer's Concern:

Converging metadynamics (MTD) in ab-initio MD is notoriously challenging. For this reason, ab-initio MD are often complemented with umbrella sampling/WHAM techniques to better evaluate the convergence of the free energy profiles. In the main text the authors state that “*The free-energy profiles of HMS formation in bulk solution and at the air-water interface were obtained by three independent metadynamics-biased AIMD simulations*”, but even averaging three 50–100 ps unconverged MTD profiles (as mentioned in the main manuscript) does not necessarily guarantee a properly converged average free energy profile.

Response: We sincerely thank the referee for the comment. We totally agree that achieving convergence in metadynamics-based AIMD simulation is inherently challenging. To address this concern, we subsequently employed the thermodynamic integration (TI)-AIMD simulation method to validate the metadynamics results for the reaction between SO_3^{2-} and HCHO in the bulk solution (**Fig. R9**). The simulation results show that our metadynamics results ($\Delta G_{\text{bulk}}^\ddagger = 5.8 \pm 0.1$ kcal/mol) is consistent with the free-energy barrier ($\Delta G_{\text{TI}}^\ddagger$) of 6.4 ± 0.7 kcal/mol by using (TI)-AIMD method, falling within the error margin.

Reviewer's Concern:

By the way, is the time serie of CV(t) in Figure 2d and 2f taken from a single MTD or is kind of averaged over multiple MTDs ?

Response: We thank the referee for the comment. The time series of CV(t) in Figure 2d and 2f are taken from a single metadynamics simulation.

Reviewer's Concern:

The lack of convergence significantly impacts the conclusions of this work. Equations 2 and 3 are used to support the claim that, under acidic conditions, the reaction is 563 times faster at the interface than in the bulk. However, these equations rely on the exponential of the free energy barriers. Given that errors of approximately 1 kcal/mol are common in such calculations that are difficult to converge, this could lead to an error of ~ 2 kcal/mol in the free energy difference in equation 3. This would reduce the calculated reaction rate enhancement from 563-fold to approximately 20-fold.

Response: We sincerely appreciate the referee for the comment. We totally agree that a precise numerical value to quantify the enhancement of the reaction rate is not sufficiently rigorous.

Consequently, we have revised the expression regarding the enhancement factor in the original manuscript. Accordingly, we have modified the original ‘563-fold’ enhancement to ‘an enhancement of *approximately two orders of magnitude*’.

Reviewer’s Concern:

The claim of a 563-fold acceleration should be presented more conservatively, and the error estimation should be discussed and performed in greater detail. For example, Figure 2e displays shaded areas as error bars, but there is no explanation of how these were computed or whether they represent a single standard deviation. Additionally, the reported error bars appear unrealistically small.

Response: We sincerely appreciate the referee for the comment. We acknowledge that the lack of a detailed description regarding the calculation of error margins in our original manuscript introduced a degree of ambiguity in the presented results. In our analysis of the metadynamics simulations, we employed cubic interpolation to estimate the error bands based on three independently obtained free-energy profiles. Indeed, we recognize that the calculated error is relatively small (0.1 kcal/mol). To reinforce the validity of our results, we further employed the thermodynamic integration (TI)-AIMD simulation method to study the $\text{HCHO} + \text{SO}_3^{2-}$ reaction in bulk solution, thereby providing an independent verification of the metadynamics-derived energy barrier (see **Fig. R9**). The simulation results show that our metadynamics results ($\Delta G_{\text{bulk}}^\ddagger = 5.8 \pm 0.1$ kcal/mol) show consistency with the free-energy barrier ($\Delta G_{\text{TI}}^\ddagger$) of 6.4 ± 0.7 kcal/mol by using (TI)-AIMD method within the error margin.

Reviewer’s Concern:

For instance, in the discussion of the reaction under basic conditions, the text states: “*The calculated activation energy in the aqueous phase is 5.8 ± 0.1 kcal/mol, which closely matches the experimental result ($\Delta G = 7.1$ kcal/mol) reported by Boyce et al.*” However, even assuming a standard deviation of 0.1 kcal/mol, an unrealistically small value, 5.8 kcal/mol is more than three standard deviations away from 7.1 kcal/mol. Clearly, this does not “closely matches the experimental result”, and the (postulated) agreement between experiments and calculations cannot be used to validate the computational methodology here adopted of averaging three (short) MTD runs.

Given its importance in ensuring reliable free energy estimates, the authors should provide additional validation or extend the simulations to achieve proper convergence.

Response: We sincerely thank the referee for the comment and suggestion. We totally agree that the statement of ‘closely matches the experimental result’ lacks sufficient rigor. As a result, we have rewritten the statement as ‘*is close to the experimental result within the relative error margin*’.

Following your suggestion, we have employed the TI-AIMD simulations to validate the metadynamics results for the reaction between SO_3^{2-} and HCHO in the bulk solution. The simulation results are presented in **Fig. R9**. For the TI-AIMD simulations, we used a cubic simulation box with dimensions $1.42 \times 1.42 \times 1.42$ nm³ containing 96 water molecules. The CV was defined as the distance between the sulfur atom of SO_3^{2-} and the carbon atom of HCHO (**Fig. R9b**). To ensure a smooth free-energy

profile, 11 sampling windows were implemented. Each window was subjected to ~ 15 ps of pre-equilibration for structural relaxation, followed by a production run of 5 ps to compute the free-energy. All simulations were performed in the canonical (NVT) ensemble at t 298.15 K, maintained via the velocity rescaling thermostat (CSVR) method. The time step for the simulation was set to 1 fs.

Fig. R9. TI-AIMD simulation of the reaction between SO_3^{2-} and HCHO in the bulk solution. (a) Gibbs free-energy profile obtained from TI-AIMD simulation with the average values of each window (blue dots) of three independent simulations, the error bar (colored orange) and the interpolation line (purple line) generated using the PCHIP method. (b) (Top) Definition of the CV for the TI-AIMD simulation. (Bottom) Snapshot structures (reactants: R_{Ti} , transition state: TS_{Ti} and product: P_{Ti}) obtained from the trajectories of TI-AIMD simulation.

As shown in **Fig. R9a**, the blue dots represent the average values from three independent simulations, and the free-energy surface (purple line) is generated by Piecewise Cubic Hermite Interpolating Polynomial (PCHIP) interpolation method. The orange error bars indicate standard deviations for each point. The simulation results show that our metadynamics results ($\Delta G_{\text{bulk}}^{\ddagger} = 5.8 \pm 0.1$ kcal/mol) show consistency with the free-energy barrier (ΔG_{Ti}^{\ddagger}) of 6.4 ± 0.7 kcal/mol by using (TI)-AIMD method within the error margin. This consistency confirms the reliability of our metadynamics-based approach for evaluating reaction energy barriers and the reaction acceleration effect at the air-water interface.

Revision:

We have revised the phrases throughout the manuscript:

‘Collectively, these effects result in approximately two orders of magnitude enhancement in heterogeneous reaction rate compared to the aqueous reaction.’ in the revised manuscript [p. 3, para. 2, lines 37–39].

‘Meanwhile, the free-energy profile of the HMS formation shows that the calculated energy barrier of $\text{SO}_3^{2-}(\text{aq.}) + \text{HCHO}(\text{aq.}) \rightarrow \text{O}^-\text{CH}_2\text{SO}_3^-$ reaction in aqueous solution is ~ 5.8 kcal/mol, is close to the experimental result (7.1 kcal/mol) within the relative error margin.’ in the revised manuscript [p. 4, para. 1, lines 1–2].

We have added a discussion in the revised manuscript:

‘Fig. 6b summarized the snapshot during the metadynamics-biased AIMD simulation for the homogeneous (top) and heterogeneous (bottom) reaction. The calculated activation energy in aqueous

phase is 5.8 ± 0.1 kcal/mol. which is close to the experimental result ($\Delta G_{exp}^{\ddagger} = 7.1$ kcal/mol) within the relative error margin. This suggests that the reaction proceeds with a moderate barrier in bulk solution. We further employed simulation using the thermal integration (TI)-AIMD simulation to validate this metadynamics result (**Supplementary Note 15 and Fig. S30**). We find that the free-energy barriers obtained from metadynamics and TI-AIMD simulations are consistent within the margin of error, thereby reinforcing the reliability of the metadynamics simulations.' in the revised manuscript [p. 15, para. 2, lines 7–14].

More discussion has been incorporated into the revised Supplementary Information [p. 53–54, Note 15, Fig. S30].

In summary, we thank the referee for the very important comments and insightful suggestions. We hope that the referee will also be satisfied with our responses about the additional discussion of sulfonate and the reliability of metadynamics simulation of our work and recommend the revised version of our manuscript for publication in *Nature Communications*.

Response to Reviewers

Reviewer #2

Reviewer's Concern:

I sincerely thank the authors for taking into account my previous comments, which I have listed below for reference:

1. *The metadynamics (MTD) simulations do not appear to have reached convergence. Since reaction rates depend exponentially on reaction barriers, the conclusions regarding accelerated rates are difficult to be supported.*
2. *Sulfonate, rather than bisulfite, is typically present at the surface of acid liquid droplets, limiting the implications of the reported findings.*
3. *Some parts of the text require better clarity.*

The authors did an excellent work on #2 and #3, also clarifying the role of sulfonate under high acid conditions in the reaction with HCHO.

Response: We thank the reviewers for the positive evaluation, particularly on #2 and #3 of the clarification of the role of sulfonate. We sincerely appreciate the time and effort the reviewer dedicated to providing constructive feedback, which has significantly strengthened the quality of our manuscript. Below, we have carefully provided a point-by-point response addressing each concern in detail.

Reviewer's Concern:

Regarding point #1, the authors acknowledged the lack of convergence in the MTD simulations, but they performed additional thermodynamic integration (TI) calculations, which support the conclusions drawn from the MTD free energy profiles.

That said, I must admit that I still have some reservations about the MTD-derived free energy profiles. For instance, the shaded areas representing error bars in Figures 2c–e appear somewhat unusual. Error bars are typically expected to be larger near the transition states (TS), which are, by definition, the least visited regions. However, in this case, the error appears larger at the minima and away from the TS.

Nonetheless, the authors have carried out thermodynamic integration to validate their findings, and they show consistency with the MTD profiles. Given the relevance and urgency of the topic for the atmospheric (but not only) physical chemistry community I agree to support this paper for Nature Communication, subject to the following minor revisions:

Response: We totally agree with the reviewer's concern regarding the shaded areas representing error bars in Figures 2c–e. We have carefully revised the manuscript according to the reviewer's concern.

Reviewer's Concern:

- a) In both the manuscript and the response letter, the authors occasionally use the term "thermal integration (TI-AIMD)", but I believe they meant "thermodynamic integration". If so, please correct it.

Response: We apologize for the mistake in the original manuscript and we sincerely appreciate the referee for the comment. We have modified the term 'thermal integration (TI-AIMD)' to '*thermodynamic integration*'.

Revision:

We have revised the phrases in the revised manuscript:

'Although the metadynamics simulations likely did not reach convergence due to the lack of recrossing events, the resulting free energy profiles and corresponding conclusions have been supported by thermodynamic integration (TI)-AIMD.' in the revised manuscript [p. 6, para. 2, lines 9–11].

'Thermodynamic integration AIMD simulations

Thermodynamic integration (TI)-AIMD simulation were carried out to investigate the reaction of $\text{HCHO} + \text{HSO}_3^-$ at the air-water interface and validate the metadynamics results for the $\text{SO}_3^{2-} + \text{HCHO}$ reaction in the bulk solution.' in the revised manuscript [p. 21, para. 1, lines 2–3].

Reviewer's Concern:

- b) Please clearly explain in the Supporting Information how the error bars were calculated.

Response: We apologize for the unclarity in the original manuscript and Supplementary Information and sincerely appreciate the referee for the comment. Following your suggestion, we have detailed the explanation of the error bar calculation both in the revised manuscript and in the revised Supplementary Information.

Revision:

We have revised the phrases in the revised manuscript:

'Relevant error band are calculated on the standard deviation by employing cubic interpolation on three independently obtained free-energy profiles. The error of the free-energy barrier is the standard deviation of the free-energy barriers of three metadynamics simulations.' [p. 5, para. 2, lines 12–14],

'Calculation of error band is same as the bulk reaction simulation.' [p. 6, para. 1, line 2],

'Calculation of error band is as same as the $\text{HCHO} + \text{HOSO}_2^-$ reaction simulation.' [p. 66, para. 1, line 5],

'Relevant error band are calculated on the standard deviation by employing cubic interpolation on three independently obtained free-energy profiles. The error of the free-energy barrier is the standard deviation of the free-energy barriers of three metadynamics simulations.' [p. 20, para. 3, lines 36–39],

'Relevant error bars of each window are calculated using the standard deviation of the corresponding free-energy values of three production runs. The free-energy profile is generated by Piecewise Cubic Hermite Interpolating Polynomial (PCHIP) interpolation of calculated free-energy values of three production for each window.' [p. 21, para. 1, lines 9–12].

and in the revised Supplementary Information:

'Relevant error band of the free-energy profile are calculated on the standard deviation by employing cubic interpolation on three independently obtained free-energy profiles. The error of the free-energy barrier is the standard deviation of the free-energy barriers of three metadynamics simulations.' [p. 4, section 2.3, lines 6–8].

'Relevant error bars (colored orange) of each window are calculated using the standard deviation of the corresponding free-energy values of three production runs. The free-energy profile (red line) is generated by Piecewise Cubic Hermite Interpolating Polynomial (PCHIP) interpolation of calculated average free-energy values of three production runs for each window.' [p. 48, caption of Supplementary Figure 27, lines 10–14].

'Relevant error bars of each window (orange) are calculated using the standard deviation of the corresponding free-energy value of three production runs. The free-energy profile is generated by Piecewise Cubic Hermite Interpolating Polynomial (PCHIP) interpolation method of calculated average free-energy values of three production runs.' [p. 52, para. 2, lines 7–10].

'Relevant error bars (colored orange) of each window are calculated using the standard deviation of the corresponding free-energy values of three production runs. The free-energy profile (red line) is generated by Piecewise Cubic Hermite Interpolating Polynomial (PCHIP) interpolation of calculated average free-energy values of three production runs for each window.' [p. 53, caption of Supplementary Figure 30, lines 3–7].

- c) Explicitly state that, although the MTD simulations likely did not reach convergence due to the lack of recrossing events, the resulting free energy profiles and corresponding conclusions have been supported by TI-AIMD. For example (but not only), on page 15, lines 12–15. This clarification will be appreciated and will enhance the credibility of the MTD results.

Response: We sincerely appreciate the referee for the comment. Following your suggestion, we have added the statement of in the revised manuscript.

Revision:

We have revised the phrases in the revised manuscript:

'Although the metadynamics simulations likely did not reach convergence due to the lack of recrossing events, the resulting free energy profiles and corresponding conclusions have been supported by

thermodynamic integration (TI)-AIMD.’ in the revised manuscript [p. 6, para. 2, lines 9–11].

‘As a result, although the metadynamics simulations likely did not reach convergence due to the lack of recrossing events, the resulting free energy profiles and corresponding conclusions have been supported by TI-AIMD.’ in the revised manuscript [p. 15, para. 2, lines 23–25].

The authors report a computational study on the formation of hydroxymethanesulfonate (HMS) from bisulfite, HSO_3^- (or sulfite, SO_3^{2-}) and formaldehyde, HCHO, at the aqueous-gas interface under acidic (or basic) conditions, respectively. As stated in the introduction, HMS is a crucial component in haze formation in polluted atmospheres. Compared to the gas-phase reaction or the reaction in homogeneous liquid water, this study reports a 563-fold enhancement of the reaction rate at the interface in acid conditions.

The authors have conducted an impressive amount of work. However, several technical and fundamental aspects are lacking or questionable, which undermines the study. Key concerns (more details are reported below):

1. The metadynamics (MTD) simulations do not appear to have reached convergence. Since reaction rates depend exponentially on reaction barriers, the conclusions regarding accelerated rates are difficult to be supported.
2. Sulfonate, rather than bisulfite, is typically present at the surface of acid liquid droplets, limiting the implications of the reported findings.
3. Some parts of the text require better clarity.

Unfortunately, based on these issues, I am sorry but I cannot support the publication of this work.

Clarity:

For future submissions, please include page and line numbers in your draft. This will help reviewers to easily reference specific parts of the manuscript.

The Abstract

The abstract is somewhat difficult to follow:

- a) “...compared to gaseous HMS formation, the *solvated reactants* ..” Does “solvated reactants” refer to those in the homogeneous bulk solution or at the air-water interface? Please clarify. Additionally, before discussing the individual contributions (i.e., polarization effects, water re-organization, etc.) to the reaction barriers, clearly state the reaction barrier values at the interface, in homogeneous bulk and gas conditions. This information is crucial to immediately grasp the catalytic effect of the heterogeneous environment.
- b) The abstract is somewhat misleading because it implies that ab-initio molecular dynamics (MD) and free energy profiles were obtained using RxDFT, while they were actually computed using BLYP-D3 DFT (as stated in the Methods section).

The Introduction

“In this atmospheric acidic condition, the dissolved S(IV) predominately exists as HSO_3^- . Therefore, this pose a contradiction that arises from the existing high concentration of HMS in the acidic aqueous environment with the abundant HSO_3^- , while the reaction rate of HCHO with HSO_3^- in solution phase is five orders of magnitude lower than that with SO^{2-} . “

This sentence is a bit unclear. I think “while” should be replaced by “because”

Bisulfite at acid aqueous environment.

The authors investigate the reactions between

- a) bisulfite (HSO_3^-) and formaldehyde (HCHO) \rightarrow HMS (acid conditions)
- b) sulfite (SO_3^{2-}) and formaldehyde (HCHO) \rightarrow HMS. (basic conditions)

in the gas phase, in bulk aqueous solution, and at the air-water interface. Reaction b) is relevant at basic conditions, Reaction a) at low pH.

Regarding Reaction a), the authors state in the abstract: “*Collectively, these factors lead to a remarkable 563-fold enhancement in the reaction rate at the air-water interface compared to the aqueous reaction.*” This is presented as one of the key findings of the paper, as seen in the abstract.

However, a recent *Nature Communications* study (Ref. 77 in the manuscript) combined XPS, Raman spectroscopy, and well-converged *ab initio* metadynamics (MTD) simulations to demonstrate that sulfonate, rather than bisulfite, is the most abundant and stable S(IV) species at the surface of an acidic air-water interface. It is surprising that, despite citing Ref. 77, the authors did not account (or at least discuss) for this finding, and focused their study on bisulfite. Indeed, in their conclusion, they state: “*Our findings reveal that, although HSO_3^- is the dominant sulfur species in acidic environments...*” However, given the evidence from Ref. 77, the conclusions about the interfacial reaction (at least for the acid conditions) starting from bisulfite and HCHO have questionable relevance and impact.

The Metadynamics

Figure 2f and Figure S3 clearly indicate that the metadynamics (MTD) simulations have not reached convergence. As shown in these figures, the collective variable (CV) exhibits only a single transition from reactants to products and never reaches a free-diffusive regime. There are no multiple recrossings, which is a necessary condition for well-converged MTD. The issue of MTD convergence has been widely discussed in the literature and across various scientific mailing lists, forum and tutorials (e.g., <https://www.plumed.org/doc-v2.9/user-doc/html/belfast-6.html>).

Converging metadynamics (MTD) in *ab-initio* MD is notoriously challenging. For this reason, *ab-initio* MD are often complemented with umbrella sampling/WHAM techniques to better evaluate the convergence of the free energy profiles. In the main text the authors state that “*The free-energy profiles of HMS formation in bulk solution and at the air-water interface were obtained by three independent metadynamics-biased AIMD simulations*”, but even averaging three 50–100 ps unconverged MTD profiles (as mentioned in the main manuscript) does not necessarily guarantee a properly converged average free energy profile. By the way, is the time series of CV(t) in Figure 2d and 2f taken from a single MTD or is kind of averaged over multiple MTDs ?

The lack of convergence significantly impacts the conclusions of this work. Equations 2 and 3 are used to support the claim that, under acidic conditions, the reaction is 563 times faster at the

interface than in the bulk. However, these equations rely on the exponential of the free energy barriers. Given that errors of approximately 1 kcal/mol are common in such calculations that are difficult to converge, this could lead to an error of ~2 kcal/mol in the free energy difference in Equation 3. This would reduce the calculated reaction rate enhancement from 563-fold to approximately 20-fold.

The claim of a 563-fold acceleration should be presented more conservatively, and the error estimation should be discussed and performed in greater detail. For example, Figure 2e displays shaded areas as error bars, but there is no explanation of how these were computed or whether they represent a single standard deviation. Additionally, the reported error bars appear unrealistically small.

For instance, in the discussion of the reaction under basic conditions, the text states: “*The calculated activation energy in the aqueous phase is 5.8 ± 0.1 kcal/mol, which closely matches the experimental result ($\Delta G = 7.1$ kcal/mol) reported by Boyce et al.*” However, even assuming a standard deviation of 0.1 kcal/mol, an unrealistically small value, 5.8 kcal/mol is more than three standard deviations away from 7.1 kcal/mol. Clearly, this does not “closely matches the experimental result”, and the (postulated) agreement between experiments and calculations cannot be used to validate the computational methodology here adopted of averaging three (short) MTD runs.

Given its importance in ensuring reliable free energy estimates, the authors should provide additional validation or extend the simulations to achieve proper convergence.

I sincerely thank the authors for taking into account my previous comments, which I have listed below for reference:

1. *The metadynamics (MTD) simulations do not appear to have reached convergence. Since reaction rates depend exponentially on reaction barriers, the conclusions regarding accelerated rates are difficult to be supported.*
2. *Sulfonate, rather than bisulfite, is typically present at the surface of acid liquid droplets, limiting the implications of the reported findings.*
3. *Some parts of the text require better clarity.*

The authors did an excellent work on #2 and #3, also clarifying the role of sulfonate under high acid conditions in the reaction with HCHO.

Regarding point #1, the authors acknowledged the lack of convergence in the MTD simulations, but they performed additional thermodynamic integration (TI) calculations, which support the conclusions drawn from the MTD free energy profiles.

That said, I must admit that I still have some reservations about the MTD-derived free energy profiles. For instance, the shaded areas representing error bars in Figures 2c–e appear somewhat unusual. Error bars are typically expected to be larger near the transition states (TS), which are, by definition, the least visited regions. However, in this case, the error appears larger at the minima and away from the TS.

Nonetheless, the authors have carried out thermodynamic integration to validate their findings, and they show consistency with the MTD profiles. Given the relevance and urgency of the topic for the atmospheric (but not only) physical chemistry community I agree to support this paper for Nature Communication, subject to the following minor revisions:

- a) In both the manuscript and the response letter, the authors occasionally use the term "thermal integration (TI-AIMD)", but I believe they meant "thermodynamic integration". If so, please correct it.
- b) Please clearly explain in the Supporting Information how the error bars were calculated.
- c) Explicitly state that, although the MTD simulations likely did not reach convergence due to the lack of recrossing events, the resulting free energy profiles and corresponding conclusions have been supported by TI-AIMD. For example (but not only), on page 15, lines 12–15. This clarification will be appreciated and will enhance the credibility of the MTD results.